# Discovering sparse transcription factor codes for cell states and state transitions during development

Leon A Furchtgott[1,2*†], Samuel Melton[1,3†], Vilas Menon[4,5], Sharad Ramanathan[1,3,4,6,7*]

[1]FAS Center for Systems Biology, Harvard University, Cambridge, United States; [2]Biophysics Program, Harvard University, Cambridge, United States; [3]Harvard Stem Cell Institute, Harvard University, Cambridge, United States; [4]Allen Institute for Brain Science, Seattle, United States; [5]Janelia Research Campus, Howard Hughes Medical Institute, Ashburn, United States; [6]Department of Molecular and Cellular Biology, Harvard University, Cambridge, United States; [7]School of Engineering and Applied Sciences, Harvard University, Cambridge, United States

*For correspondence: leon.
furchtgott@gmail.com (LAF);
sharad@cgr.harvard.edu (SR)

†These authors contributed
equally to this work

Competing interests: The
authors declare that no
competing interests exist.

Reviewing editor: Nir Yosef,
University of California, United
States

**Abstract** Computational analysis of gene expression to determine both the sequence of lineage choices made by multipotent cells and to identify the genes influencing these decisions is challenging. Here we discover a pattern in the expression levels of a sparse subset of genes among cell types in B- and T-cell developmental lineages that correlates with developmental topologies. We develop a statistical framework using this pattern to simultaneously infer lineage transitions and the genes that determine these relationships. We use this technique to reconstruct the early hematopoietic and intestinal developmental trees. We extend this framework to analyze single-cell RNA-seq data from early human cortical development, inferring a neocortical-hindbrain split in early progenitor cells and the key genes that could control this lineage decision. Our work allows us to simultaneously infer both the identity and lineage of cell types as well as a small set of key genes whose expression patterns reflect these relationships.

## Introduction

During development, pluripotent cells make a series of lineage decisions to give rise to the different cell types of the body. These lineage decisions are controlled by intra-cellular molecular networks that include transcription factors and signaling molecules. There are two fundamental challenges associated with understanding the differentiation of individual cells. The first is to identify lineage relationships: how cells and their progeny move from pluripotent through intermediate to terminally differentiated cell states. The second is to identify the key molecular drivers that allow cells to make fate decisions along their developmental trajectory.

Reconstructing cell lineages has traditionally involved prospectively tracking cells and their progeny using a variety of imaging or genetic tools (*Buckingham and Meilhac, 2011*; *Frumkin et al., 2008*; *Orkin and Zon, 2008*; *Sulston et al., 1983*). Recent progress in single-cell sequencing techniques (*Grün et al., 2015*; *Jaitin et al., 2014*; *Macosko et al., 2015*; *Patel et al., 2014*; *Paul et al., 2015*; *Treutlein et al., 2014*; *Zeisel et al., 2015*) allows for a complementary view of the transcriptional states of individual cells during the course of development, providing static snapshots of the dynamics of the underlying molecular network. But inferring lineage relationships and the dynamics of the underlying molecular networks has proved difficult using transcriptional data alone, in part because of the high dimensional nature of these data. Overcoming this challenge would be

particularly useful for understanding the development of human organs such as the brain where traditional lineage tracing experiments are more difficult.

High dimensional data analysis techniques such as PCA, ICA, or t-SNE, are useful at reducing dimensionality. However, since the resulting axes represent linear combinations of a large number of features (for example, expression levels of each gene), interpreting the analysis or making experimental predictions is sometimes challenging. Meanwhile, traditional statistical methods such as linear multivariate regression have limited applicability for detecting patterns in high dimensional data (*Advani and Ganguli, 2016*; *Donoho and Tanner, 2009*). The challenges inherent in high-dimensional data analysis such as identifying discriminatory features are further exacerbated as the fraction of relevant features decreases (*Donoho and Tanner, 2009*). Computational techniques currently in use to cluster single cell data or to infer relationships among cells are built on these approaches, thereby assuming that all high-variance genes are equally relevant for pattern detection (*Marco et al., 2014*; *Satija et al., 2015*; *Trapnell et al., 2014*). In contrast, decades of work in developmental biology have revealed that combinations of a few transcription factors can be sufficient to experimentally perturb cell fate and developmental decisions (*Gilbert, 2014*; *Graf and Enver, 2009*; *Takahashi and Yamanaka, 2006*) suggesting that the expression patterns of a few genes may be most relevant for making computational inferences. Therefore, there is a need to detect patterns involving a small fraction of all genes. Unfortunately, except in the case of well-studied lineage decisions, we do not know the identity of this fraction.

In statistics, techniques relying on L1 regularization have been successful in contexts where the number of informative variables is known to be small but whose identities are unknown, both for regression problems (*Baraniuk, 2007*; *Candès et al., 2006*; *Tibshirani, 1996*; *Wainwright, 2009*) and for clustering (*Witten and Tibshirani, 2010*). Inspired by these successes in statistics, our aim here is to discover generalizable sparse patterns in gene expression data during development (if they exist), and to exploit these patterns to computationally infer the dynamics of cell state transitions from high-dimensional transcriptional data obtained during the course of development.

In this manuscript we analyze expression patterns among cell types with known lineage relationships in late hematopoiesis and discover a pattern in a sparse subset of genes that correlates with these relationships. We develop a Bayesian framework based on this gene expression pattern to simultaneously infer lineage transitions and the key genes that drive them. We apply this method to reconstruct the lineage tree among a different set of cell types in early hematopoietic development, and in this process identify many known drivers of early hematopoiesis, including *Gata1*, *Cebpa* and *Ebf1*. We further extend our method to analyze single-cell gene expression data, using genes exhibiting the discovered pattern to cluster cells from early brain development and to infer lineage relationships between these clusters. Our analysis reveals a split from early progenitors to putative neocortex and mid/hindbrain cell types, as evidenced by the mutually exclusive expression of region-specific genes such as *FOXG1*, *LHX1*, and *POU3F2* (*BRN2*). This prediction was validated experimentally in a separate work (*Yao et al., 2017*). We finally discuss the advantages of using sparse patterns for making inferences and for modeling the underlying gene regulatory networks.

## Results

### Discovering sparse patterns correlated with lineage transitions

In order to identify gene expression patterns that are robustly predictive of lineage relationships, we analyzed gene expression data from 41 cell types during B- and T- cell development that have an experimentally established developmental lineage (*Figure 1A*, *Heng et al., 2008*). We searched for sparse patterns of gene expression amongst groups of three cell types from this collection; subsets of three are the minimal set in which measures of relative similarity can be used infer relative lineage relationships.

We identified 150 triplets of cell types with experimentally verified lineage relationships from B- and T- cell development (*Heng et al., 2008*) (*Figure 1—source data 1*). Three such triplets are shown in *Figure 1A*. These triplets constituted both cell fate decisions (for example, cell type A gives rise to cell type B and C) and lineage progressions (cell type B gives rise to cell type A which then gives rise to cell type C). For each triplet, we noted which cell type was the progenitor or intermediate cell type ('root' cell type A) and which cell types were not ('leaf' cell types B and C). Note

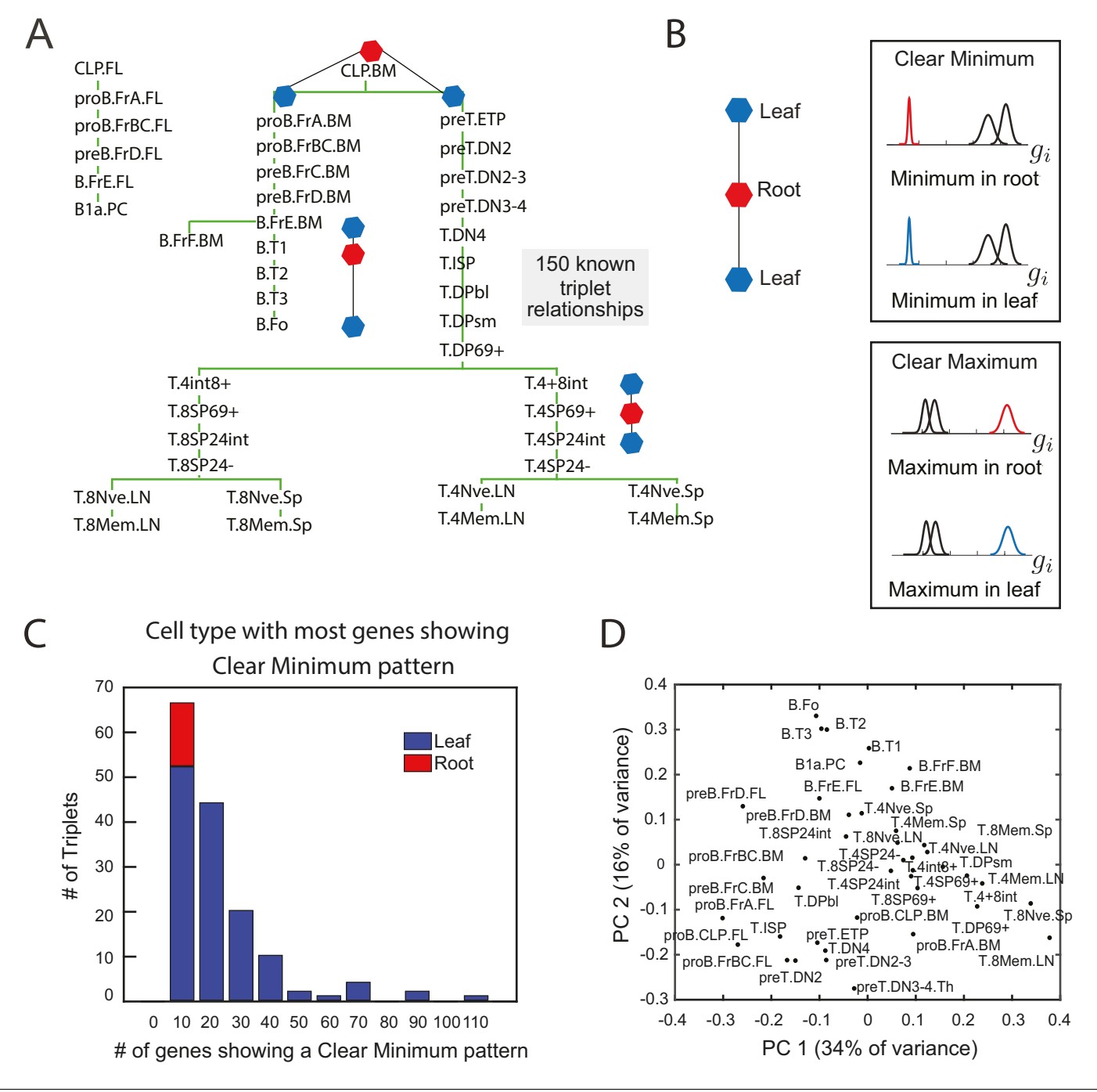

**Figure 1.** The clear minimum pattern is robustly detected in leaf cell types throughout triplet lineages in B- and T-cell development. (**A**) Developmental lineage tree showing relationships among 41 cell types in B- and T-cell development (*Heng et al., 2008*). Three triplets – the minimal subset of the tree from which relative distances can be studied – are denoted, each including an intermediate root cell type (red) and terminal leaf cell types (blue). 150 triplets among all sets of three cell types within five steps on the lineage tree were extracted for pattern-detection. (**B**) For each triplet of cell types (left), each gene's expression level can have the clear minimum pattern in either the root or the leaves (right, top box) where the distribution of gene expression levels in one cell type is well separated from the other two (left, p<0.005 in a two sample t-test); clear maximum pattern (left, bottom) in the root or the leaves: the gene has a clear maximum in one of the cell types (right, p<0.005 in a two sample t-test). (**C**) A histogram of the number of genes showing the clear minimum pattern among the 150 triplets with known developmental topology. Triplets in which the root has the most genes showing the pattern shown in red; triplets in which one of the leaves has the most genes showing the pattern shown in blue. None of the triplets with more than 10 genes showing the pattern have the most genes with a clear minimum in the root (no red in any histogram bar except for the
*Figure 1 continued on next page*

*Figure 1 continued*

left most)(D) Principal component analysis of microarray data from the cell types in B- and T- cell development does not reflect known lineage relationships in (A).

The following source data and figure supplements are available for figure 1:

**Source data 1.** Triplets used for pattern detection.
**Source data 2.** List of mouse transcription factors.
**Figure supplement 1.** Clear minimum and clear maximum patterns.
**Figure supplement 2.** The clear minimum pattern is observed across different types of triplets.
**Figure supplement 3.** Distinction between related and unrelated triplets.

that even in the case in which cell type B gives rise to cell type A which gives rise to cell type C, we will refer to A as the 'root' and B and C as 'leaves' for that particular triplet. We analyzed transcription factor gene expression data for these triplets from the Immunological Genome Consortium (*Heng et al., 2008*) since transcription factors are the ultimate drivers of cell fate decisions.

Surprisingly, we found that specific expression patterns involving only single genes correlated well with lineage relationships between three cell types. Genes that are not differentially expressed within a triplet of cell types convey no information about relationships between the cell types. Therefore, we select genes with expression variability among the three cell types. The expression pattern of such genes can belong to one of only two possible patterns: (*Figure 1B*): the *clear minimum pattern*: the gene has a clear minimum level in one of the three cell types, with its distribution of gene expression levels being well-separated from the other two (p<0.005 in both two sample *t*-tests between the minimum cell type and the other two cell types; *Figure 1—figure supplement 1A*); the *clear maximum pattern*: the gene has a clear maximum level in one of the cell types (p<0.005 in both sample *t*-tests; *Figure 1—figure supplement 1B*). Note that both patterns can be satisfied simultaneously if the distribution of expression levels in the three cell types are all well-separated with a clear maximum and minimum. We tested if either of the two patterns correlated with the lineage topologies between the three cell types with known lineage relationships (*Figure 1C*, *Figure 1—figure supplement 1C*).

Triplets of cell types can be separated into categories based on how many genes exhibit the aforementioned patterns. 56% of the triplets contained more than 10 genes exhibiting the clear minimum pattern, and in 100% of these triplets the majority of genes with expression fitting this pattern reached their minimum expression in one of the leaves (*Figure 1C*) and never in the root of the triplet. The frequency with which the pattern correctly indicated the lineage relationship increased with the number of genes within a triplet exhibiting the pattern, thus suggesting a confidence measure. The genes showing the clear minimum pattern fell into two distinct groups, corresponding to whether the minimum expression level was in one or the other of the two leaves (*Figure 1—figure supplement 1A*; *Figure 1—figure supplement 2C*). Thus the expression pattern of the total set of clear minimum genes correlated with the topology. Since genes showing the clear minimum pattern correlated with lineage relationships (*Figure 1C*) between cell states both in the case of branches and linear sequences of cell state transitions, we refer to them as *transition genes*.

We further verified that the clear minimum pattern could be observed (a) in the set of all 25,194 genes (*Figure 1—figure supplement 2A*), (b) using FDR-adjusted p-values (*Benjamini and Hochberg, 1995*) (*Figure 1—figure supplement 2B*), (c) in triplets of different lengths along the lineage tree (*Figure 1—figure supplement 2D*), (d) in triplets both containing only internal nodes and including terminal nodes (*Figure 1—figure supplement 2E*), (e) and in both lineage progression and cell fate decision triplets (*Figure 1—figure supplement 2F*).

The clear maximum pattern was a poorer indicator of lineage relationships (*Figure 1—figure supplement 1C*). 83% of the triplets had more than 10 genes exhibiting this pattern, but 10% of those showed the majority of genes with expression fitting this pattern reaching their maximum in the root

while the others did so in the leaves. Crucially, the integrity of the relationship between the clear maximum pattern and lineage topology was not correlated with the number of genes exhibiting the pattern (*Figure 1—figure supplement 1C*). While the clear maximum pattern did not correlate with lineage relationships, genes exhibiting this pattern identify individual cell types, and therefore we will refer to them as *marker genes*.

There are many examples of genes known to be functionally important for lineage decisions whose expression patterns fit the clear minimum pattern. In the case of lateral inhibition commonly used during development, progenitor cells express genes together (for example, *Notch* and *Delta*) which are differentially expressed in the differentiated states (only *Notch* or only *Delta*) (*Perrimon et al., 2012*) reaching the minimum expression level in one of the leaves. The same pattern is also seen in multiple examples of lineage decisions often involving mutual inhibition, where key genes expressed in the progenitor are differentially regulated in the progeny (*Graf and Enver, 2009*; *Qi et al., 2013*; *Thomson et al., 2011*; *Zhang et al., 1999*). In each of these cases, key genes reach minimal expression levels in one of the leaves of the triplets.

The observation that genes exhibiting the clear minimum pattern are correlated with the lineage topology of a triplet of cell types further revealed that (i) only this fraction of transcription factors can be useful for inferring lineage relationships, and (ii) the identity of this fraction depends on which group of three cell types were analyzed. As the subset of genes that are informative varies based on the triplet of cell types being considered, establishing lineage relationship between all cell types at once as opposed to three at a time could be challenging. Indeed, our attempt to reconstruct the lineage relationships between all cell types using methods based on PCA failed (*Figure 1D*).

We further evaluated the clear minimum pattern in 100 triplets in which there was no clear relation between the cell types (*Figure 1—source data 1*). We found that while there were a substantial number of genes exhibiting a clear minimum in one of the three cell types in unrelated triplets, their minima were evenly distributed amongst the cell types. To quantify that the minima were evenly distributed, we counted the fraction of genes $f_i$ which reached a clear minimum in cell type $i = A, B, C$ (for A, B, and C unrelated cell types), and for each triplet, we computed the entropy $S = -\sum_{i=A,B,C} f_i \log(f_i)$. We compared the distribution of the entropy and the number of genes showing a minimum in any triplet for unrelated and related triplets (*Figure 1—figure supplement 3A*). The unrelated triplets have higher entropy and typically more genes with a minimum level. This suggested that unrelated triplets show a distinct pattern from the related triplets.

## Using patterns to infer lineages

Together, these observations suggested a strategy for inferring the lineage topology between three cell types: each gene showing the clear minimum pattern with a minimum expression level in a particular cell type increases to the probability that this cell type is not the root of the topology (*Figure 2A*). We next developed a statistical machinery to systematically detect this pattern in gene expression data and to use the resulting sparse subset of genes to infer lineage relationships between three cell types at a time. We then used the inferred relationships between all sets of three cell types as constraints to determine the full developmental lineage tree.

The classifications of genes as transition or marker genes in *Figure 1* were based on p<0.005 in a two-sample *t*-test. To implement such classifications probabilistically without arbitrary cutoffs we developed a statistical framework to infer the lineage relationships between each set of three cell types A, B, and C and find the key sets of transition genes (those genes that show the clear minimum pattern), given gene expression data $\left\{g_i^{A,B,C}\right\}$ in those cell types. We determined the probability of any possible topological relationship between the cell types $T = \{A, B, C, \emptyset\}$ referring to cell type A, B, or C being the root of the triplet, or $\emptyset$ which corresponds to the case where the data does not suggest any lineage relationship between the three cell types because either no significant pattern could be detected, or multiple genes exhibiting the minimum pattern suggested conflicting topologies. Rather than an absolute classification of genes as showing a pattern or not, we calculated the probability $p\left(\alpha_i = 1 \mid g_i^{A,B,C}\right)$ of each gene $i$ being a marker gene (denoted by $\alpha_i = 1$), i.e. gene $i$ showing the clear maximum pattern, and the probability $p\left(\beta_i = 1 \mid g_i^{A,B,C}\right)$ of it being a transition gene denoted by $\beta_i = 1$, i.e. gene $i$ showing the clear minimum pattern (Materials and methods).

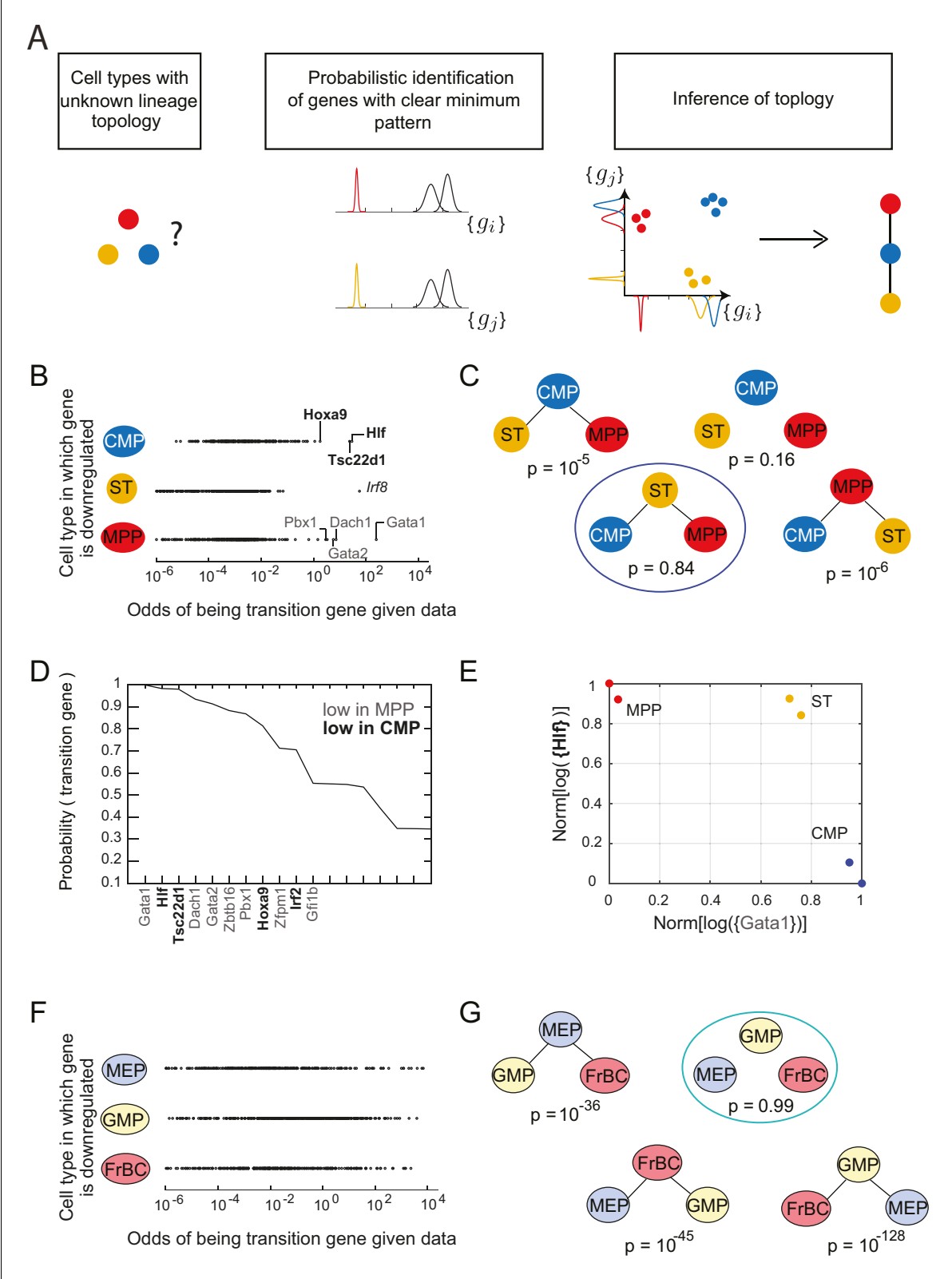

**Figure 2.** Identification of topology and transition genes (showing clear minimum pattern) for each triplet of cell types. (**A**) Schematic for the statistical inference of lineage topology for 3 cell types. Genes with a clear minimum pattern indicate which cell types that are not the root (see **Figure 1C**) and hence allow inference of the topological relationship. (**B**) Dot plot (each dot representing a gene) of the cell type that is most likely to have the minimum mean expression of each gene among CMP, ST and MPP as a function of the odds $\mathcal{O}_i$ of that gene being a transition gene. Each gene votes

*Figure 2 continued on next page*

*Figure 2 continued*

against the topology whose root has the minimum mean among the three cell types, and this vote is weighted by the odds that the gene is a transition gene (*Equation 1*). Two groups of genes, labeled by their names, have high odds of being transition genes and thus cast a strong vote against CMP or MPP being the root.(C) The computed probability of the topology given gene expression data indicates 0.84 probability that ST is the intermediate type.(D) The plot of the probabilities of genes being transition genes for triplet ST/MPP/CMP, given gene expression data and that the topology is MPP – ST – CMP. The names of the 10 genes with the highest probability of being transition genes are shown. Probabilities are calculated assuming the prior odds $\frac{p(\beta_i=1)}{p(\beta_i=0)}$ = 0.05 (see main text). There are two classes of transition genes: one for which gene expression in CMP is greater than expression in MPP (regular font), and another for which gene expression in MPP is greater than expression in CMP (bold font).(E) Plot of the replicates of ST, MPP and CMP in the gene-expression space of the two classes of transition genes (with probability > 0.8). Plotted on each axis is the mean normalized log expression level of the transition genes in the class, each class is denoted in curly brackets by the name of the transition gene with the highest probability.(F) Dot plot for triplet MEP/GMP/FrBC of the cell type that is most likely to have the minimum mean expression as a function of the odds $O_i$ of that gene being a transition gene.(G) The computed probability of the topology given gene expression data is the null hypothesis (p=0.99).

The following source data and figure supplements are available for figure 2:

**Source data 1.** Early hematopoietic cell types considered.
**Source data 2.** Probabilities of topologies for triplets of hematopoietic cell types.
**Source data 3.** Probabilities of transition and marker genes for the hematopoietic lineage tree.
**Figure supplement 1.** Probability of topology depends on prior odds.
**Figure supplement 2.** Plots of the length of the triplets distinguishing the traditional model and in the Adolfsson model.

We derived the probability of a given topology $T$ given the expression data as (Materials and methods):

$$p\left(T \mid \left\{g_i^{A,B,C}\right\}\right) \propto \prod_i \left(1 + \frac{3}{2}\mathcal{O}_i\left[1 - p\left(\mu_T^i \text{ is min} \mid g_i^{A,B,C}\right)\right]\right), \tag{1}$$

where, $\mathcal{O}_i = p\left(\beta_i = 1 \mid g_i^{A,B,C}\right)/p\left(\beta_i = 0 \mid g_i^{A,B,C}\right)$ is the odds of gene $i$ being a transition gene and thus having a unique minimum. The term $p\left(\mu_T^i \text{ is min} \mid g_i^{A,B,C}\right)$ is the probability that the mean $\mu_T^i$ of the distribution of the expression levels of gene i in the root cell type $T$ is less than the mean in the other two cell types. The odds implicitly contains the only free parameter in our analysis, the prior odds $\frac{p(\beta_i=1)}{p(\beta_i=0)}$, which defines the number of genes we expect to show the clear minimum pattern *a priori*, and functions as a sparsity parameter for the inference. Qualitatively, in the above equation, every gene casts a vote $-p\left(\mu_T^i \text{ is min} \mid g_i^{A,B,C}\right)$ against the cell type $T$ in which its mean expression is minimal being the root. Further, this vote is weighted by the odds $\mathcal{O}_i$ of gene $i$ being a transition gene. Thus, genes with a clearer minimum pattern get larger votes in determining which cell type is not the root. In practice, these quantities are computed numerically (Materials and methods).

We note further that if a substantial number of genes cast votes against each of the cell types, then the probability of the null topology ∅ increases. We computed the probability of obtaining the null topology among the 150 related triplets and 100 unrelated triplets from our training set. The distribution of the probability of obtaining the null topology was considerably different between the related triplets and the unrelated triplets, with an AUC of 0.96 (*Figure 1—figure supplement 3B–C*).

## Application to hematopoietic gene expression data

We used our statistical framework to recreate the lineage of early hematopoietic differentiation. We considered 11 early hematopoietic progenitors from the ImmGen Consortium microarray data set (*Heng et al., 2008*) (*Figure 2—source data 1*). These cell types and their associated relationships were not included in the data set used earlier to study the correlations of the two patterns and lineage topologies. Several features of the early hematopoietic lineage tree are debated

(*Adolfsson et al., 2005*; *Iwasaki and Akashi, 2007*) (*Figure 2—figure supplement 2A*). Given only the gene expression data for these different subpopulations of cells, we determined the lineage relationships and the key factors associated with each lineage decision. We calculated the probabilities of topology and marker and transition genes for the $\binom{11}{3} = 165$ possible triplets of cell types using our statistical framework (*Figure 2—source data 2*). To illustrate our method, we first described the analysis of the expression data from two such triplets of cell types: CMP/ST/MPP and MEP/GMP/FrBC (*Figure 2B–G*). We then assembled the triplets to form an undirected lineage tree (*Figure 3*; *Video 1*).

Following *Equation 1*, each gene votes against the topology whose central node has the minimum expression of that gene among the three cell types, and this vote is weighted by the odds that the gene is a transition gene. To illustrate this for the triplet of cell types CMP, ST and MPP, we plotted the topology each gene voted most against, *i.e.* the topology $T$ for which $p\left(\mu_T^i \text{ is min} \mid g_i^{CMP,ST,MPP}\right)$ is the maximum, versus the odds $\mathcal{O}_i$ of that gene being a transition gene (*Figure 2B*).

We find two groups of genes that are much more likely to be transition genes than any of the other genes, with values of $O_i \sim 10^2$ compared to $10^0$ at most for other genes (*Figure 2B*, regular and bold fonts). These two groups of genes have a large value for either $p\left(\mu_{CMP}^i \text{ is min} \mid g_i^{CMP,ST,MPP}\right)$ or $p\left(\mu_{MPP}^i \text{ is min} \mid g_i^{CMP,ST,MPP}\right)$ and thus vote against $T = \textbf{CMP}$ (cell type CMP is the intermediate) or against $T = \textbf{MPP}$ (cell type MPP is the intermediate). Together these genes that have a high odds of being transition genes appear to most support topology $T = \textbf{ST} \equiv CMP - ST - MPP$.

In fact, the intuition in *Figure 2B* is borne out in the calculation of $p\left(T \mid \left\{g_i^{CMP,ST,MPP}\right\}\right)$. Using *Equation 1* above and assuming a sparsity parameter of 0.05, we calculate that there is an 84% chance that the topology is $\textbf{ST}$ (*Figure 2C*; *Figure 2—figure supplement 2B*). Although gene *Irf8* (*Figure 2B*, italic font) is strongly downregulated in ST and is expressed at higher levels in CMP and MPP (*Figure 2C*), we note that its signal is overwhelmed by the large number of genes downregulated in either CMP or MPP, illustrating the statistical nature of the framework.

For each triplet, we evaluated each gene's probability of being a transition or marker gene (*Figure 2—source data 3*). *Figure 2D* shows the names and associated probabilities of the 12 genes most likely to be transition genes for the triplet $\text{CMP} - \textbf{ST} - \text{MPP}$. The transition genes fall into two groups, corresponding to the two groups in *Figure 2B*. One group, which includes genes *Gata1*, *Dach1*, and *Gata2*, has higher expression in CMP than in MPP; the other group, which includes *Hlf*, *Tsc22d1*, and *Hoxa9*, has higher expression in MPP. Although the values of the probabilities of the genes being transition genes vary with the value of the sparsity parameter, the relative order of different genes does not change. The genes identified include many genes previously identified as being important for lineage specification (*Crispino, 2005*; *Gazit et al., 2013*; *Miyawaki et al., 2015*). The transition genes we discovered thus not only have gene expression patterns that reflect the lineage decision but also include functionally important genes.

In addition to the transition genes, we identified marker genes ($p(\alpha_i = 1 \mid \{g_i\}, T) > 0.8$) present only in ST (including *Mpl* and *Rai14*, consistent with [*Solar et al., 1998*]) and then symmetrically downregulated in both CMP and MPP (*Figure 2—figure supplement 1C*). Marker genes for CMP include *Srf*, *Zeb2*, *Rbpj* and *Irf8* (consistent with [*Goossens et al., 2011*; *Kurotaki et al., 2013*; *Ragu et al., 2010*; *Robert-Moreno et al., 2005*; *Tamura et al., 2000*]); marker genes for MPP include *Satb1*, consistent with (*Satoh et al., 2013*). Although these genes were not used to determine the topology, they are good markers for cell types ST, CMP and MPP.

Plotting the cell types using the mean expression levels of the two transition gene class captures the fork in the gene expression space associated with the cell-fate decision (*Figure 2E*). In contrast with the PCA analysis of the cell types (*Figure 2—figure supplement 1E*), in which MPP appears to be an intermediate between the hematopoietic stem cell types (LT and ST) and CMP, the projection of the cell types onto the transition-gene subspace shows that ST splits into CMP and MPP.

In contrast to the case of the triplet of cell types CMP/ST/MPP, for triplet MEP/GMP/FrBC, the distributions of genes supporting different topologies are similar (*Figure 2F*). Thus the most likely topology calculated using *Equation 1* is the null hypothesis (99%), which is that transition genes, if

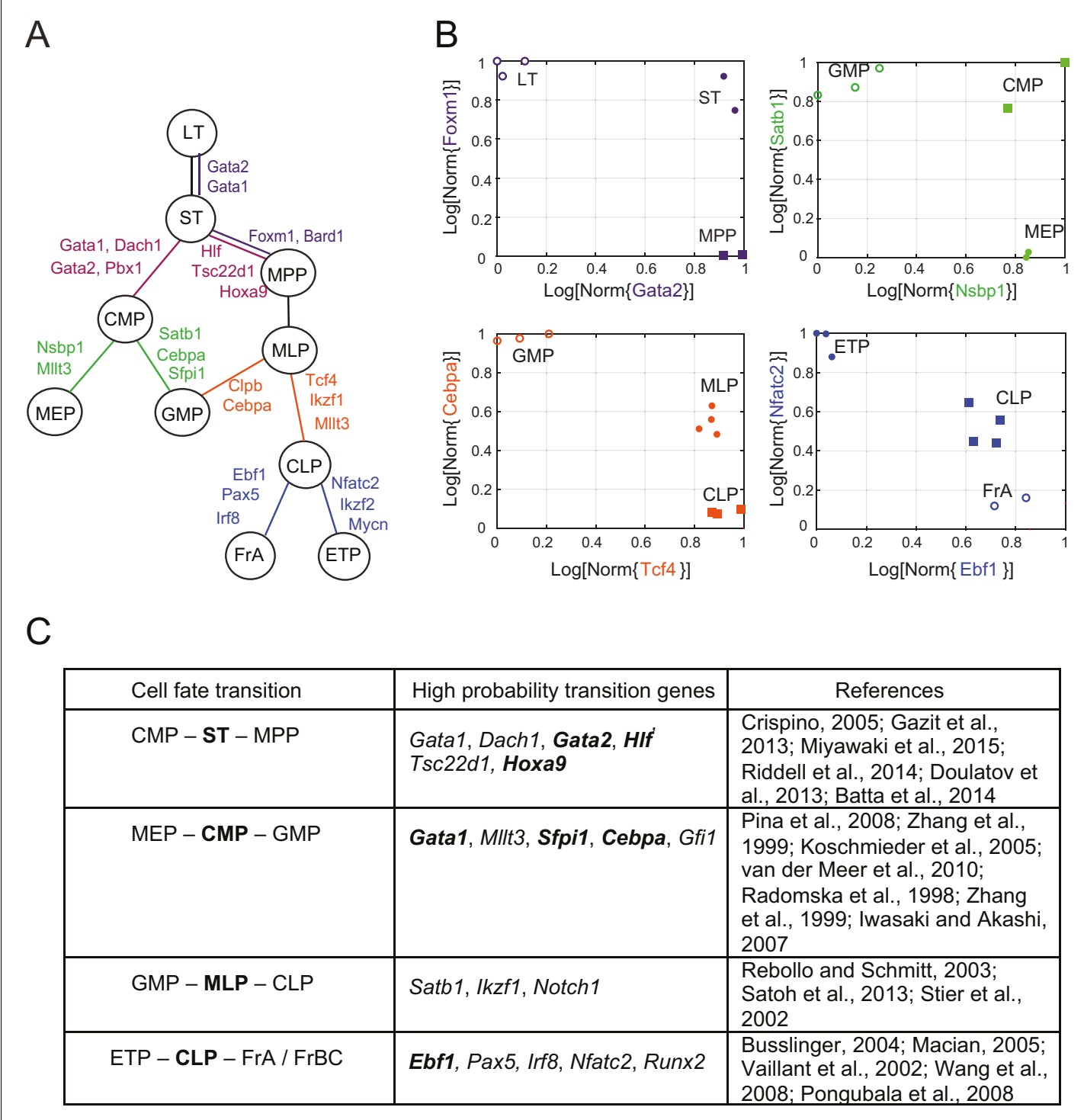

**Figure 3.** Reconstruction of lineage tree and key transition gene for early hematopoiesis. (A) Final lineage tree, recapitulating the inferred triplet topologies, with top inferred transition genes indicated along cell fate decisions. (B) Plot of the replicates of different cell types in the gene-expression space of the transition gene classes (probability > 0.8) for 4 cell-fate transitions along the inferred lineage tree in (A). Plotted on each axis is the mean normalized log expression level of the transition genes in the class. The axis labels and data points are color-coded according to the colors in (A). (C) Table with selected transition genes for early hematopoietic cell-fate transitions, along with references to published validations of their functional role. Genes known to be effective for reprogramming are shown in bold.

*Figure 3 continued on next page*

*Figure 3 continued*

The following source data and figure supplements are available for figure 3:

**Source data 1.** Marker genes for early hematopoiesis.
**Figure supplement 1.** Reconstruction of lineage tree from individual triplets.
**Figure supplement 2.** Inferred lineage tree and transition genes for intestinal development.

they exist, do not have patterns that depend on the cellular topology (*Figure 2G*, *Figure 2—figure supplement 1D*), in which case there is insufficient evidence to classify the triplet according to a particular non-null topology. The distribution of the maximal probability of non-null topologies in the different triplets is heavily concentrated near 1, allowing for clear separation between null and non-null triplets (*Figure 2—figure supplement 1E*). Null topologies were identified by the algorithm for triplets with cells that are from three terminal nodes (for example, the triplet MEP/GMP/FrBC) or from triplets that contain very distantly related triplets (for example, LT/CLP/ETP) (*Figure 2—source data 2*).

## A lineage tree for early hematopoiesis

We next reconstructed the early hematopoietic lineage tree and identified transition genes involved in the different cell state transitions using all the non-null triplet relationships as constraints on the lineage relationships between all cell types. Out of the 165 possible triplets of hematopoietic progenitors, 144 showed one single non-null topology with probability greater than 0.6 over a range of the prior odds from $10^{-6}$ to $10^2$. We next determined an undirected graph that recapitulates all of the individual triplet topologies (note that we are only inferring triplet topologies and are not inferring directionality). For example, although triplet CMP/LT/MPP has topology CMP – **LT** – MPP (*Figure 3—figure supplement 1A*), we could determine that LT cannot be the *direct* progenitor of CMP or MPP, because ST is an intermediate between LT and both cell types (*Figure 3—figure supplement 1B–C*). We could thus 'prune' this triplet when inferring the full graph (*Figure 3—figure supplement 1D–E*). A visualization of the pruning process is shown in *Video 1*, where successive triplets are added to the graph, creating new edges and pruning others, leading to the final undirected tree. In practice, though, the pruning process was performed on all triplets simultaneously, not in succession.

The ST/CMP/MPP triplet (*Figure 2B–E*) immediately distinguishes between two competing models regarding the hierarchy of early hematopoietic progenitors. According to the traditional picture (*Iwasaki and Akashi, 2007*), MPP is the progenitor of CMP, and ST is the progenitor of MPP – therefore MPP should be an intermediate between ST and CMP and the topology of triplet ST/MPP/CMP should be ST –

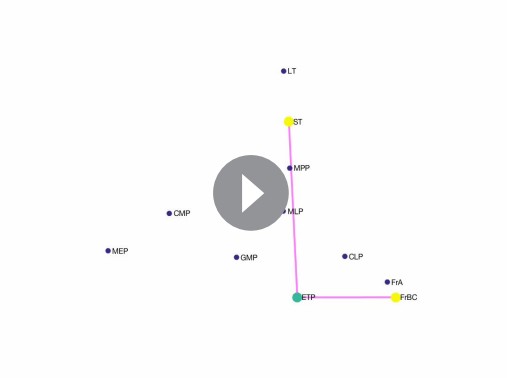

**Video 1.** Tree-building process for early hematopoietic lineage Animation of the triplet assembly and pruning process for reconstructing the early hematopoietic lineage. For illustrative purposes, triplets (with p>0.6) are successively selected at random (in practice, the assembly and pruning process was performed on all triplets simultaneously; the resulting tree does not depend on the order in which the triplets are selected). The nodes of the current triplet are highlighted in yellow; if a topology is recognized for the triplet, the root is shown in green and the leaves in yellow, and the triplet edges are shown in magenta. If adding the triplet causes another triplet to be pruned, the soon-to-be-pruned (i.e. offending) edge is highlighted in red. The resulting pruned graph is then shown before adding the next triplet. As more triplets are considered, more edges between nodes are added and then pruned, leading to the final tree.

MPP – CMP (*Figure 2—figure supplement 1A*, left). According to a model suggested by Adolfsson and colleagues (*Adolfsson et al., 2005*), ST splits into CMP and MPP (*Figure 2—figure supplement 1A*, right), and the topology should be CMP – ST – MPP. We identify both CMP – **ST** – MPP and CMP – **LT** – MPP as the correct topologies, lending support to the Adolfsson model. The Adolfsson and traditional models differ in the topology of 9 triplets. The inferred expected topologies of 8 out of these nine triplets support the Adolfsson model, which led to the identification of the final tree (*Figure 2—figure supplement 2*).

The lineage tree that we determined is consistent with the Adolfsson model and contains three additional lineage decisions (*Figure 3A–B*). First, CMP gives rise to MEP (megakaryocyte/erythroid progenitor) and GMP (granulocyte/macrophage progenitor). Second, MPP gives rise to MLP (multilineage progenitor), which then splits into the GMP and CLP (common lymphoid progenitor) cell types. In the final lineage decision, CLP gives rise to the ETP (pre-T) and FrA (pre-pro-B) cell types.

For each triplet of cell types along the tree, we identified transition and marker classes of genes. Among the 14 triplets that contained only adjacent cell types, we identified on average 24 marker genes per cell type and 25 transition genes (probability threshold of 0.8, prior odds $p(\beta_i = 1)/p(\beta_i = 0) = 0.05$). Many genes we discovered as belonging with high probability to the transition and marker classes of genes at each lineage decision are known in the literature to be functionally important genes, including classic hematopoietic regulators such as *Cebpa*, *Sfpi1*, *Gata1*, *Satb1*, *Irf8* and *Ebf1* (see full tables with references in *Figure 3C* and *Figure 3—source data 1*). Additionally, the genes identified include many genes successfully used in hematopoietic reprogramming experiments, including *Gata2* and *Pbx1* (*Figure 3C*). Together these observations suggest that the sparse subspace of transition and marker genes identified by our framework not only allows for accurate reconstruction of the lineage hierarchy but also constitutes a set of candidates for relevant biological functions.

As further validation of the inference method, we compared it to the method proposed by Grun et al. on a single-cell intestinal development data set (*Grün et al., 2015*, *2016*). We inferred lineages between each cell type based on their cluster identifications, excluding clusters with fewer than 10 cells, and constructed an undirected lineage tree by taking triplets with probability > 0.6 and applying the pruning rule (*Figure 3—figure supplement 2A*). The only disagreement between the two methods is the progression from crypt base columnar cells ($C_2$) to different populations of Goblet cells ($C_4$ and $C_8$). Grun et al. hypothesize a $C_2$ – $C_8$ – $C_4$ progression, while we infer the triplet $C_8$ – $C_2$ – $C_4$ with p>0.99, suggesting that the progenitor $C_2$ gives rise to both differentiated Goblet subpopulations. Both lineage trees are supported by the literature (*van der Flier and Clevers, 2009*). The high probability transition genes included many factors well known for their roles in tissue homeostasis and development (*Figure 3—figure supplement 2B*), notably *Klf4* (*Yu et al., 2012*), *Atoh1* (*VanDussen and Samuelson, 2010*), *Spdef* (*Noah et al., 2010*), *Foxa1/Foxa2* (*Ye and Kaestner, 2009*), and *Tcf3* (*Merrill et al., 2001*).

## Inferred lineage tree for human excitatory neuronal progenitors from in vitro single-cell data over 80 days of differentiation

The ease with which single-cell transcriptomic data can be generated (*Grün et al., 2015*; *Jaitin et al., 2014*; *Macosko et al., 2015*; *Patel et al., 2014*; *Paul et al., 2015*; *Treutlein et al., 2014*; *Zeisel et al., 2015*) presents an opportunity to understand the dynamics of the underlying networks that lead individual cells to their final fate. We studied the differentiation of stem cells both into germ layer progenitors (*Jang et al., 2017*) and into cortical neurons. To study the latter, we analyzed single-cell gene expression data from 2217 cells from an in vitro differentiation protocol for early human neuronal development (*Yao et al., 2017*). Briefly, human embryonic stem cells (hESCs) were subjected to a SMAD inhibition-based cortical induction phase, a progenitor expansion phase, and a neural differentiation phase. Single cells were sorted at 12, 26, 54, and 80 days into differentiation, and their gene expression was profiled using the SMART-Seq2 technique (*Picelli et al., 2013*). In the initial clustering of the single-cell data, dimensionality reduction by PCA (into 15 above-noise components) followed by t-SNE (*Van Der Maaten and Hinton, 2008*; *Satija et al., 2015*) showed separation by day and *SOX2* expression (*Figure 4A*). However, the number of predicted clusters varied depending on the perplexity parameter (*Figure 4—figure supplement 1A–B*). In addition, no clear lineage or distance relationship among the putative types is immediately apparent from this clustering. Analysis of this data with other recent methods such as Monocle and

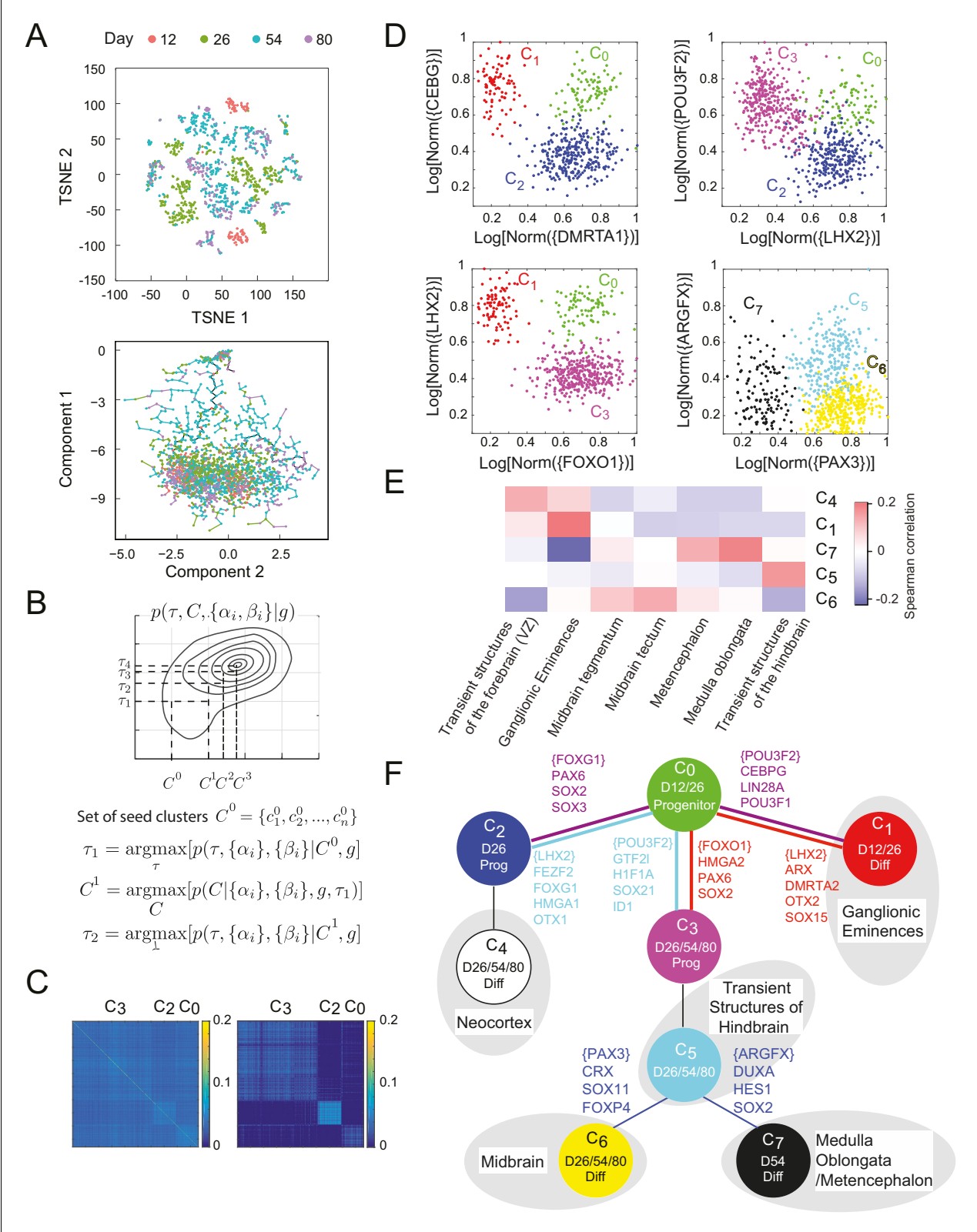

**Figure 4.** Inference of lineage tree and key transitions genes using single cell expression data from in vitro differentiated developing human brain. (**A**) RNA-seq data from single cells collected at days 12, 26, 54, and 80 from a human brain in vitro differentiation protocol (*Yao et al., 2017*) were analyzed using a variety of existing methods. Partitioning single-cells into cell types through non-linear dimensionality reduction using t-SNE (top) depends on the perplexity parameter (set here to 5, see *Figure 4 – Figure Supplement 1A-B*) and does not allow for mechanistic understanding. Independent

*Figure 4 continued on next page*

*Figure 4 continued*

component analysis of all transcription factors with Monocle (bottom) does not show clear structure and could not inform reconstruction of lineage relationships. (B) Maximization algorithm to determine most likely cluster identities $\{C\} \equiv \{c_1, c_2, \ldots, c_n\}$, sets of transitions $\{T\}$, marker genes ($\alpha_i = 1$) and transition genes ($\beta_i = 1$), given single-cell gene expression data $\{g_i\}$. Starting from a seed clustering scheme $\{C_0\}$, iterative maximization of the conditional probabilities $p(\{T\}, \{\alpha_i\}, \{\beta_i\}|\{g_i\}, \{C\})$ and $p(\{C\}|\{g_i\}, \{T\}, \{\alpha_i\}, \{\beta_i\})$ converges to most likely set $(\{C\}, \{T\}, \{\alpha_i\}, \{\beta_i\})$ (C) Cell-cell covariance matrix between cells using only the associated high probability marker and transition genes show the final cluster assignments $c_0, c_2$ and $c_3$ (right) in contrast to using all transcription factors (left). (D) Selected high probability triplets of clusters plotted in the axes defined by two sets of transition gene classes for each triplet. $c_1 - c_0 - c_2$ (top right, $p(T = c_0 \mid \{g_i^{c_0,c_1,c_2}\}) > 0.99$), plotted in transition gene class $\{CEBPG\}$ also including *POU3F1, POU3F2, NR2F1, NR2F2, ARX, LIN28A, TOX3, ZBTB20, PROX1* and *SOX15*, and class $\{DMRTA1\}$ also including *HES1, HES5, FOXG1, PAX6, HMGA2, SOX2, SOX3, SOX9, SOX6, SP8, OTX2, TGIF, ID4, TCF7L2*, and *TCFL1*. $c_2 - c_0 - c_3$ (top left, $p(T = c_0 \mid \{g_i^{c_0,c_2,c_3}\}) = 0.96$), plotted in transition gene class $\{LHX2\}$ also including *FEZF2, FOXG1, HMGA1, SP8, OTX1, SOX11, GLI3, SIX3, ETV5*, and class $\{POU3F2\}$ also including *GTF2I, HIF1A, ID1, ID3, PROX1, SALL1, SOX21, TCF12, TRPS1, ZHX2*. $c_1 - c_0 - c_3$ (bottom left, $p(T = c_0 \mid \{g_i^{c_0,c_1,c_3}\}) > 0.99$), plotted in transition gene class $\{FOXO1\}$ also including *HMGA2, PAX6*, and *SOX2*, and class $\{LHX2\}$ also including *DMRTA2, HMGA1, ARX, LIN28A, OTX2, LITAF, NANOG, POU3F1, SOX15*. $c_6 - c_5 - c_7$ (bottom right, $p(T = c_5 \mid \{g_i^{c_5,c_6,c_7}\}) > 0.99$), plotted in transition gene class $\{PAX3\}$ also including *CRX, SOX11, EBF2, FOXP4, ASCL1, FOXO3*, and *SIX3*, and class $\{ARGFX\}$ also including *DUXA, HES1, NFIB, PPARA, SOX2, SOX7*, and *SOX9*. (E) Correlations between differentiated cell clusters (Figure 4 – Figure Supplement 4D) and bulk population samples from brain regions (*in vivo* developmental human data) (*Miller et al., 2014*). Neuronal cell types can be identified with specific spatial regions of the brain to interpret the topology of the lineage tree. Expression signatures of *SOX2*+ cell types $c_0, c_2$ and $c_3$ were dominated by pluripotency factors, and are not shown. (F) Inferred lineage tree for brain development. Genes associated with neocortical development, and mid-/hind-brain progenitors, and specific neuronal cell types are identified as high probability transition genes and are corroborated by mapping information from in vivo data. Clusters color-coded similarly to (D). D12/26/54/80 labels indicate time of collection of cells within each cell type. Prog refers to *SOX2*+ cells, Diff refers to *SOX2-/DCX*+ cells (*Figure 4—figure supplement 1C–D*).

The following source data and figure supplements are available for figure 4:

**Source data 1.** Final cluster identities of single cells from in vitro cortical differentiation.
**Source data 2.** Probabilities of topologies for triplets of single-cell clusters.
**Source data 3.** Probabilities of Transition and Marker Genes for the Human Brain Developmental Lineage Tree.
**Source data 4.** Human Brain Development SmartSeq2 Census.
**Source data 5.** List of Human Transcription Factors.
**Figure supplement 1.** Cluster identity and sparse coding in neuronal differentiation.
**Figure supplement 2.** A selection of recent lineage-determination methods for single cell transcriptomic analysis applied to an in vitro neuronal differentiation data set (*Yao et al., 2017*).

Monocle2 (*Trapnell et al., 2014*), TSCAN (*Ji and Ji, 2016*), and StemID (*Grün et al., 2016*) did not clearly reconstruct lineage or infer key genes regulating transitions (*Figure 4A* – Bottom, *Figure 4— figure supplement 2*). Monocle2 (*Figure 4—figure supplement 2A*) produces a tree with complex branching, but *SOX2*+ progenitors and *DCX*+ differentiated neurons do not clearly separate.

Analyzing data from single-cell profiling presents an additional challenge relative to data from population-level profiling because cell types are not previously known and must be inferred from the data. Computationally, it is necessary define a measure of similarity in the gene expression profiles of individual cells so as to be able to cluster them and define cell states. Here again, it is necessary to identify the correct gene subspace to use for clustering. Clustering and determining lineage have typically been performed sequentially and treated as independent problems (*Satija et al., 2015*; *Trapnell, 2015*; *Trapnell et al., 2014*). However, we found that it is informative to solve both these problems simultaneously.

In our framework, the relevant feature set for clustering is the set of marker and transition transcription factors from the triplets with non-null topologies. But determination of these sets of genes and of the transition topologies depends itself on knowledge of the cluster identities. Following previous work on sparse clustering (*Witten and Tibshirani, 2010*), we simultaneously determined

optimal clusters, lineage topology and sets of transition and marker genes by iteratively selecting transition and marker genes and clustering the data using this set of features (**Figure 4B**).

In order to utilize information about developmental distances in clustering, we iteratively maximized a joint distribution, $P(T, \{C\}^m | \{g_i\})$, over the developmental tree, $T$, and the set of clusters of cells, $\{C\}^m = \{c_1^m, c_2^m, \ldots, c_K^m\}$ (**Figure 4B**). Starting with a clustering $\{C\}^m$, we first inferred the set of genes $\{\alpha_i = 1\}$ and $\{\beta_i = 1\}$ which were identified in high probability triplets, $p(T | \{g_i\}, \{C\}^m) > 0.6$, and we then re-clustered in this new subspace to obtain the clusters for the next iteration $\{C\}^{m+1}$.

Initial analysis based on the gap statistic (**Tibshirani et al., 2001**) suggested that the single-cell gene expression profiles clustered into 20 clusters of cell types. We chose a seed $\{C\}^0$ for the iterated clustering procedure by intentionally over-clustering the data into 40 clusters using spectral K-medoids. By being overly discriminative in our initial clustering, we ensured that all genes with differential regulation would be classified as either marker genes or transition genes, and would be preserved in later clustering iterations. We iterated the clustering-inference procedure until the dimension of the re-clustering subspace changed by less than 10% of the total transcription factor space. In this resulting subspace of 469 genes, we finally clustered the cells into the final configuration $\{C\}^f = \{c_0, c_1, \ldots, c_7\}$ using K-medoids (**Figure 4C**, **Figure 4—source data 1**) where the number of clusters K = 8 was chosen based on the gap statistic (**Tibshirani et al., 2001**).

We inferred 45 high probability triplets ($p(T | \{g_i\}) > 0.6$) between the final cell clusters (**Figure 4— source data 2**). Four such triplets are shown in **Figure 4D**, plotted in axes defined by transition genes for each triplet (**Figure 4—source data 3**). Starting with progenitor cell states (*SOX2+*, **Figure 4—figure supplement 1C**), we manually appended cell clusters to the tree according to their time information and in agreement with inferred topological restrictions. The first transition involves the production of day 12 neuronal cell type $c_1$ from day 12 progenitor $c_0$, which is observed in the triplets $c_1 - c_0 - c_2$ and $c_1 - c_0 - c_3$. The $c_1 - c_0 - c_2$ triplet ($p(T = c_0 | \{g_i^{c_0, c_1, c_2}\}) = 0.99$, **Figure 4D** – top left) is mediated by 47 transition genes between $c_0$ and $c_1$ and 87 between $c_0$ and $c_2$ ($p(\beta_i = 1 | \{g_i\}, T) > 0.8$). Transition genes expressed highly in the $c_1 - c_0$ branch include *CEBPG*, *POU3F1*, *POU3F2*, *NR2F1*, *NR2F2*, *ARX*, *LIN28A*, *TOX3*, *ZBTB20*, *PROX1*, and *SOX15* which have been previously implicated in proliferation of forebrain progenitors (**Au et al., 2013**; **Borello et al., 2014**; **Cimadamore et al., 2013**; **Dominguez et al., 2013**; **Yang et al., 2015**) and the migratory behaviors of ganglionic eminences (**Kanatani et al., 2008**; **Kessaris et al., 2014**; **Lodato et al., 2014**; **Olivetti and Noebels, 2012**; **Reinchisi et al., 2012**). The $c_0 - c_2$ transition genes include *DMRTA1*, *HES1*, *HES5*, *FOXG1*, *PAX6*, *HMGA2*, *SOX2*, *SOX3*, *SOX9*, *SOX6*, *SP8*, *OTX2*, *TGIF*, *ID4*, *SOX3*, *TCF7L2*, and *TCFL1* which are known to be expressed in forebrain progenitors of the developing neocortex, thalamus, and hypothalamus (**Abraham et al., 2013**; **Pozniak et al., 2010**; **Shimojo et al., 2011**; **Tzeng and de Vellis, 1998**; **Wang et al., 2006**), and are known to establish dorsal forebrain regional identity; (**Azim et al., 2009**; **Bani-Yaghoub et al., 2006**; **Borello et al., 2014**; **Gaston-Massuet et al., 2016**; **Hagey et al., 2014**; **Hutton and Pevny, 2011**; **Johansson et al., 2013**; **Kikkawa et al., 2013**; **Kishi et al., 2012**; **Manuel et al., 2011**; **Miyoshi and Fishell, 2012**; **Ohtsuka et al., 2001**; **Ross et al., 2003**; **Shen and Walsh, 2005**; **Sur and Rubenstein, 2005**; **Yang et al., 2015**; **Zembrzycki et al., 2007**).

The progenitor cell types form the triplet $c_2 - c_0 - c_3$ ($p(T = c_0 | \{g_i^{c_0, c_2, c_3}\}) = 0.96$, **Figure 4D** – top right). The $c_2 - c_0$ branch is mediated by 39 transition genes including *LHX2*, *FEZF2*, *FOXG1*, *HMGA1*, *SP8*, *OTX1*, *SOX11*, *GLI3*, *SIX3*, and *ETV5* which suggest that $c_2$ is comprised of cortical progenitors (**Appolloni et al., 2008**; **Greig et al., 2013**; **Kishi et al., 2012**; **Manuel et al., 2011**; **Raciti et al., 2013**). The $c_0 - c_3$ branch is mediated by 55 transition genes including *GTF2I*, *HIF1A*, *ID1*, *ID3*, *PROX1*, *POU3F2*, *SALL1*, *SOX21 TCF12*, *TRPS1* and *ZHX2*, which are associated with mesencephalon and metencephalon regional development, as well as having known involvement with midbrain/hindbrain organizer identity (**Buck et al., 2001**; **Enkhmandakh et al., 2009**; **Inoue et al., 2012**; **Jaegle et al., 2003**; **Kunath et al., 2002**; **Lavado and Oliver, 2007**; **Milosevic et al., 2007**; **Ohba et al., 2004**; **Uittenbogaard and Chiaramello, 2002**; **Yao et al., 2017**). We additionally inferred the triplet $c_2 - c_0 - c_3$ ($p(T = c_0 | \{g_i^{c_0, c_1, c_3}\}) > 0.99$, **Figure 4D** – bottom left), which suggests a three way split from early progenitor $c_0$ into early differentiated neuron $c_1$, and progenitors $c_2$ and $c_3$.

The continuation of the $c_2$ branch is inferred through triplet $c_0 - c_2 - c_4$ ($p(T = c_2 | \{g_i^{c_0, c_2, c_4}\}) = 0.97$). The $c_2 - c_4$ branch includes transition genes *BCL11A*, *EMX2*, *FOXP2*

and *RORB*, which are known to be associated with the neocortex and neuronal identity (*Cánovas et al., 2015*; *Ebisu et al., 2016*; *Greig et al., 2016*; *Jabaudon et al., 2012*; *Wiegreffe et al., 2015*; *Woodworth et al., 2016*; *Zembrzycki et al., 2007*).The triplet $c_0 - c_3 - c_5$ ($p(T = c_3 \mid \{g_i^{c_0,c_3,c_5}\}) > 0.99$) meanwhile is characterized by transition genes in the $c_3 - c_5$ branch including *ASCL1, FOXP2, PAX3, POU3F4, ZIC1, ZIC4, HOXB2*, and *EN2*, which have been shown to regulate fate acquisition in the midbrain/hindbrain (*Agoston et al., 2012*; *Ang, 2006*; *Di Bonito et al., 2013*; *Elsen et al., 2008*; *Hegarty et al., 2013*; *Miller et al., 2011*; *Tan et al., 2014*).

The $c_5$ cell cluster differentiates into two distinct clusters of post-mitotic neurons – $c_6$ and $c_7$ ($p(T = c_5 \mid \{g_i^{c_5,c_6,c_7}\}) > 0.99$, *Figure 4D* – bottom right). The $c_5 - c_6$ branch is inferred from transition genes including *PAX3, CRX, SOX11, EBF2, FOXP4, ASCL1, FOXO3*, and *SIX3* which have strong expression in developing dopaminergic and gabaergic neurons of the midbrain (*Agoston et al., 2012*; *Erickson et al., 2010*; *Pino et al., 2014*; *Yang et al., 2015*; *Yin et al., 2009*; *Zhang et al., 2002*). Transition genes in the $c_5 - c_7$ branch include *ARGFX, DUXA, HES1, NFIB, PPARA, SOX2, SOX7*, and *SOX9*, which are known to be associated with the medulla oblongata region of the hindbrain (*Fawcett and Klymkowsky, 2004*; *Kameda et al., 2011*; *Kumbasar et al., 2009*; *Madissoon et al., 2016*; *Matsui et al., 2000*; *Stolt et al., 2003*).

In addition to interpreting these individual transition genes defining the major branch splits, we correlated the expression over all predicted transition and marker genes in neuronal clusters to in vivo developmental human data (*Miller et al., 2014*). The in vivo data comprise a representative range of microarray data sampled from different parts of the developing brain at post-conception week 15, including forebrain proliferative regions, midbrain, and hindbrain. The differences in data acquisition methods (RNAseq vs. microarray, single-cell vs heterogeneous populations) resulted in relatively low correlations overall, but there are clear associations between individual clusters and specific brain regions (*Figure 4E*). Specifically, $c_1$, maps to the ganglionic eminences, suggesting an interneuron identity, whereas $c_5$, $c_6$ and $c_7$ show better mapping to mid- and hindbrain structures, and $c_4$ appears to be more closely related to neocortex. Overall, this global comparison, combined with the identification of genes with known regional expression, suggests that the inferred clusters from the in vitro data capture the diversity of differentiation into the early stages of the major neuronal lineages (*Figure 4F*). These lineage predictions based on our analysis techniques were verified experimentally using viral barcoding in a separate work (*Yao et al., 2017*).

To estimate the sparsity of the underlying network and to find a minimal subset of genes through which lineage could be inferred, we replicated the analysis while only considering a limited set of genes per triplet. We assembled a collection of 20 triplets with maximal leaf-to-leaf distance of 4 nodes, and non-null inferred topology. For each triplet, we ranked genes based on their odds of being a transition gene, $\mathcal{O}_i$, agnostic of the true topology of the triplet. We then replicated the inference process using only the N genes with the greatest odds (*Figure 4—figure supplement 1E*). We found that with as few as 4 genes per triplet, the correct lineage topology could be inferred for all of the triplets. Genes with greatest odds comprise a restricted subset of genes for further experimental investigation. Further, these findings suggest that the dynamics of expression of just four specific genes are sufficient to monitor a particular lineage decision in single cells.

## Discussion

Finding an informative subspace of variables for data analysis is a general problem in machine learning, both for regression and clustering (*Tibshirani, 1996*; *Witten and Tibshirani, 2010*); the innovation in this paper is to use a statistical pattern learned from known biology to inform this subspace search. The approach we take here is complementary to methods that project expression variability onto coordinates of PCA, ICA or t-SNE maps (*Marco et al., 2014*; *Satija et al., 2015*; *Trapnell et al., 2014*), which are combinations of all variables. Searching for sparse representation of the dynamics has the advantage of providing interpretability and experimental direction (*McGibbon and Pande, 2017*). Following the dynamics of this small set of high-probability transition genes via fluorescent tagging could allow for the tracking of lineage decisions of individual cells in real time. Further, these genes provide a list of candidates for drivers of fate decisions, and hence a set of experimental hypotheses.

Not all genes give us equal information about the dynamics of differentiation. We discovered that genes showing the clear minimum pattern are most predictive of the sequence of lineage transitions

during development. Although our pattern discovery and subsequent lineage reconstruction does not assume any functional role for the clear minimum pattern, we note that this pattern is shown by genes known to be regulators of development during hematopoiesis. The same pattern observed in many differentiating systems (*Graf and Enver, 2009*; *Qi et al., 2013*; *Thomson et al., 2011*; *Zhang et al., 1999*), and is consistent with mutual inhibition. Mutual inhibition, in turn, is hypothesized to play an important role in maintaining multi-stable systems and in mediating transitions between different stable states of multi-stable systems (*Ferrell, 2012*).

Discovery of sparse representations of the cell states and variability between them demonstrates the efficacy of low dimensional descriptions of the system. Understanding the dynamics and transitions of complex physical systems composed of a large number of variables has been driven by the discovery of low dimensional order parameters (*Anderson, 1978*; *Landau and Lifshitz, 1951*). As opposed to measuring and modeling states as high dimensional objects in their native representation, order parameters provide low dimensional descriptions of the states and dynamics, which has proven crucial in developing both qualitative and quantitative models. Finding small subsets of genes which captures the lineage transitions in cells analogously provides a low dimensional subspace that captures the dynamics in genetic networks and can be useful for modeling (*Jang et al., 2017*). An accompanying paper allows us to exploit this idea to extract mathematical models for the underlying molecular circuits from single cell gene expression data obtained during germ layer differentiation (*Jang et al., 2017*).

## Materials and methods

### In vitro neuronal differentiation

Single-cell transcriptomic data from the in vitro neural differentiation procedure was obtained as described in *Yao et al. (2017)* (Supplemental information):

hESCs were dissociated with Accutase and plated on Matrigel-coated 24-well plates at $2.5 \times 105$ cells/cm2 in DMEM/F12 (#11330–032), $1 \times$ N2, $1 \times$ B27 without vitamin A, 2 mM Glutamax, 100 µM non-essential amino acids, 0.5 mg/mL BSA, 1X Pen-Strep, and 100 µM 2-mercaptoethanol (referred to as basal medium; all from Thermo Fisher, Waltham, MA) with 20 ng/mL FGF2 (Thermo Fisher) and 2 µM thiazovivin. Cortical induction was initiated by changing to the basal medium with 5 µM SB431542 (StemRD, Burlingame, CA), 50 nM LDN193189 (Reagents Direct, Encinitas, CA) and 1 µM cyclopamine (Stemgent, Lexington, MA) (referred to as NIM). NIM was changed daily for 11 days. On day 12, cells were dissociated and seeded on Matrigelcoated 24-well plates at $5 \times 105$/cm2 in basal medium with 20 ng/mL FGF2 and 2 µM thiazovivin. Progenitor expansion was initiated on D13 by changing to serum-free human neural stem cell culture medium (NSCM, #A10509–01 from Thermo Fisher) containing 20 ng/mL FGF2 and 20 ng/mL EGF. NSCM was changed daily for 6 days. Cultures were passaged once more on D19 with Accutase and replated at $5 \times 105$ cells/cm2. On D26, cells were dissociated with Accutase and seeded on 24-well plates sequentially coated with poly-D-lysine (Millipore, Billerica, MA) and laminin (Thermo Fisher) at $1 \times 105$ cells/cm2 in basal medium supplemented with 20 ng/mL FGF2 and 2 µM thiazovivin. On D27, medium was changed to a 1:1 mixture of DMEM/F12 and Neurobasal medium (#21103–049) supplemented with 100 µM cAMP (Sigma-Aldrich, St Louis, MO), 10 ng/mL BDNF (R and D Systems, Minneapolis, MN), 10 ng/mL GDNF (R and D Systems) and 10 ng/mL NT-3 (R and D Systems) (referred to as ND). Cells were maintained in ND medium for four weeks until day 54 with half medium change every other day. Quality of differentiations was routinely assessed by immunostaining at D12 (*PAX6* and *DCX*), at D26 (*LHX2*, *SOX2*, *EOMES*, *POU3F2*, and *TBR1*), and at D54 (*MAP2* costained with *TBR1*, *CTIP2*, *SATB2*). In addition flow cytometry at D26 (*EOMES*, *SOX2* and *PAX6*) was performed. Typically, *EOMES* at day 26 proved the most valuable quality control metric (~10% of cells by both flow cytometry and immunostaining) and predicted failure at D54. Specifically, when *EOMES* was low (<1% of cells) differentiations failed and were typically dominated by *POU3F2*+ cell types and/or non-neural 'other' cell types. These failed differentiations were eliminated from further analysis, typically ~20% of experiments (5 of 19 experiments in 2016). 50 bp paired-end Smart-Seq2 libraries were prepared from these cells as previously described (*Picelli et al., 2013*) and mapped as described in *Thomsen et al., 2016* and *Yao et al. (2017)*. As each cell was profiled independently (without pooling before amplification, as in methods such as Cel-Seq, STRT, or Drop-Seq), we did not observe the

batch effects present in pooling-based methods. Although cells were profiled in plates (batches), there was no significant plate-related variation - this Fixed single-cell transcriptomic characteriza is most likely due to amplification being carried out separately in each well of the plate, thereby precluding cross-talk among barcodes, or plate-related differences in amplification. This is clear from the mixing of cells from different plates in the clustering. A complete census is provided (*Figure 4—source data 4*).

## Gene expression data

Hematopoietic gene expression data were downloaded from the Immunological Genome Project (*Heng et al., 2008*); GEO GSE15907) and log-2 transformed. We restricted the genes considered to 1459 transcription factors. Brain development expression data were log-2 transformed, and 1460 transcription factors were individually normalized by dividing by the 90th-percentile expression value. Lists of mouse and human transcription factors are provided (*Figure 1—source data 2*, *Figure 4—source data 5*).

## Software

Calculations were performed using custom written MATLAB code (The Mathworks) on the Harvard Research Computing Odyssey cluster. Code is available at https://github.com/furchtgott/sibilant. t-SNE was done using the package provided in (*Van Der Maaten, 2009*). Monocole was run in R (*Core Team, 2015*; *Trapnell et al., 2014*).

## Description of algorithm

The algorithm proceeds according to the following steps:

1. Find initial seed clustering configuration $\{C\}^0$ using K-medoids where K is chosen to be larger than the cluster number inferred from the Gap statistic (*Tibshirani et al., 2001*)
2. For all triplets of clusters, find most likely $T$ and $\{\alpha_i\}$ and $\{\beta_i\}$ given $\{C\}^m$:
   a. Compute $p\left(g_i^{A,B,C} \mid T, \alpha_i, \beta_i, \{C\}^m\right)$ by integrating numerically over $p(\mu, \sigma)$ (*Equations 6, 9, and 10*).
   b. Compute $p\left(T, \{\alpha_i\}, \{\beta_i\} \mid \left\{g_i^{A,B,C}\right\}, \{C\}^m\right)$ using *Equations 20 and 29*.
   c. Identify mostly likely topology $T$ and set of $\{\alpha_i\}$ and $\{\beta_i\}$.
3. Recluster $\{g\}$ using K-medoids in the space of all $\{\alpha_i\}$ and $\{\beta_i\}$ for the triplets with probability $p\left(T \mid \left\{g_i^{A,B,C}\right\}, \{C\}^m\right) > 0.6$ of being non-null. Determine new clustering configuration $\{C\}^{m+1}$.
4. Repeat steps 1 and 2 until convergence of $\{C\}$.
5. Determine the most likely tree connecting cell clusters, recapitulating high-probability triplet topologies.

Each of these steps is described in the following section.

## Bayesian framework for inferring cluster identities, state transitions, and marker and transition genes simultaneously

### Notation; Bayes' rule

Given gene expression data from single cells $\{g_i\}$, we built a probabilistic framework to simultaneously infer cell cluster identities, $\{C\} \equiv \{c_A, c_B, \ldots\}$, the sequence of transitions $T$ between these clusters, the key sets of marker genes $\{\alpha_i\}$ that define each cell cluster, and genes $\{\beta_i\}$ that determine the sequence of transitions between clusters. We maximized the joint probability distribution of these variables given the gene expression data, $p(T, \{c_A, c_B, \ldots\}, \{\alpha_i\}, \{\beta_i\} \mid \{g_i\})$ to determine the maximum likelihood estimates of these parameters.

We first consider how to solve this problem in the case in which there are three cell clusters, and we will later build a tree using all possible combinations of three cell clusters. Let the set of three cell clusters be $c_A$, $c_B$ and $c_C$ with gene expression data $\left\{g_i^{A,B,C}\right\}$ for all genes ($i = 1$ to $N$) and all cells. The term $g_i^{A,B,C}$ denotes the expression data for just gene $i$ in cells in clusters $c_A$, $c_B$, and $c_C$. The topology $T$ of the relationships between cell clusters $c_A$, $c_B$ and $c_C$ can take on four possible values: $T = \mathcal{A}$: cell cluster $c_A$ is in the middle (either $c_A$ is the progenitor of $c_B$ and $c_C$, or $c_A$ is an intermediate cell type between $c_B$ and $c_C$); $T = \mathcal{B}$: cell cluster $c_B$ is in the middle; $T = \mathcal{C}$: cell cluster $c_C$ is

in the middle; or $T = \emptyset$: we cannot determine the topology. Complementarily, for each gene $i$ we define variables $\alpha_i$ and $\beta_i$, where $\alpha_i = 1$ and $\beta_i = 0$ if the gene is a marker gene, $\alpha_i = 0$ and $\beta_i = 1$ if the gene is a transition gene, and $\alpha_i = \beta_i = 0$ otherwise. Our task is to determine the probability $p\left(T, \{c_A, c_B, \ldots\}, \{\alpha_i\}, \{\beta_i\} \mid \left\{g_i^{A,B,C}\right\}\right)$ given gene expression data for all genes $\left\{g_i^{A,B,C}\right\}$.

According to Bayes' rule, $p\left(T, \{c_A, c_B, \ldots\}, \{\alpha_i\}, \{\beta_i\} \mid \left\{g_i^{A,B,C}\right\}\right)$ is proportional to the probability of the gene expression data given $T$, $\{C\}$, $\{\alpha_i\}$ and $\{\beta_i\}$:

$$p\left(T, \{c_A, c_B, \ldots\}, \{\alpha_i\}, \{\beta_i\} \mid \left\{g_i^{A,B,C}\right\}\right)$$
$$= \frac{p\left(\left\{g_i^{A,B,C}\right\} \mid T, \{C\}, \{\alpha_i\}, \{\beta_i\}\right) p(\{\alpha_i\}, \{\beta_i\} \mid T, \{C\}) \, p(T|\{C\}) \, p(\{C\})}{p\left(\left\{g_i^{A,B,C}\right\}\right)} \quad (2)$$

The denominator of the right hand side of **Equation 2** is a normalization constant. Expressions $p(\{\alpha_i\}, \{\beta_i\} \mid T, \{C\})$, $p(T|\{C\})$, and $p(\{C\})$ are respectively the prior probabilities of $\{\alpha_i\}$ and $\{\beta_i\}$ given $T$ and $\{C\}$, the prior probability of $T$ given $\{C\}$, and the prior probability of $\{C\}$. We assume that in the absence of any expression data, the probability that a gene is a transition or marker gene is independent of the topology and clustering configuration: $p(\{\alpha_i\}, \{\beta_i\} \mid T, \{C\}) \, p(T|\{C\}) \, p(\{C\}) = p(\{C\}) p(T) \prod_i p(\alpha_i, \beta_i)$, and $p(T) = 1/4$ .

## Conditional independence

In our model, we assume that knowing the clustering configuration $\{C\}$, the topology $T$ and whether or not a gene is a marker or transition gene is sufficient to determine the probability distribution for its expression levels in each of the cell clusters. Therefore, the gene expression patterns of different genes are conditionally independent given the topology, clustering and gene type:

$$p\left(\left\{g_i^{A,B,C}\right\} \mid T, \{C\}, \{\alpha_i\}, \{\beta_i\}\right) = \prod_i p\left(g_i^{A,B,C} \mid T, \{C\}, \alpha_i, \beta_i\right) \quad (3)$$

Thus, **Equation 2** becomes

$$p\left(T, \{c_A, c_B, \ldots\}, \{\alpha_i\}, \{\beta_i\} \mid \left\{g_i^{A,B,C}\right\}\right) = \frac{p(\{C\}) p(T) \prod_i p\left(g_i^{A,B,C} \mid T, \{C\}, \alpha_i, \beta_i\right) p(\alpha_i, \beta_i)}{p\left(\left\{g_i^{A,B,C}\right\}\right)} \quad (4)$$

We maximize the evaluated $p\left(T, \{c_A, c_B, \ldots\}, \{\alpha_i\}, \{\beta_i\} \mid \left\{g_i^{A,B,C}\right\}\right)$ with respect to $T$, $\{C\}$ and each of the $\alpha_i$ and $\beta_i$ to obtain the most likely relationships between cell types $c_A$, $c_B$ and $c_C$, as well as the genes most likely to be marker and transition genes.

## Expression for $p\left(g_i^{A,B,C} \mid T, \{C\}, \alpha_i = 0, \ \beta_i = 1\right)$ (transition genes)

To infer $p\left(T, \{c_A, c_B, \ldots\}, \{\alpha_i\}, \{\beta_i\} \mid \left\{g_i^{A,B,C}\right\}\right)$, we need a model to compute the probability of the gene expression data for each gene, $p\left(g_i^{A,B,C} \mid T, \{C\}, \alpha_i, \beta_i\right)$, given $T$, $\{C\}, \alpha_i$ and $\beta_i$, following **Equation 4**. Our model for the probability distribution of the expression of a single transition gene $i$ in the three cell types $p\left(g_i^{A,B,C} \mid T, \{C\}, \alpha_i = 0, \beta_i = 1\right)$ is defined solely by the geometry of the arrangement of the cell types in gene expression space, as described in the main text. For example, for $T = \mathcal{A}$ and $\beta_i = 1$, our model is that the distribution of the expression levels of $g_i^{A,B,C}$ in the three cell types A, B and C has the smallest mean value in either B or C but not in A (**Figure 2A**). If the distribution of the expression of gene $i$ in cell type A is $D_A\left(g_i^A \mid \mu_A^i, \sigma_A^i, \{C\}\right)$ (we assume a log-normal distribution) with a mean $\mu_A^i$ and standard deviation $\sigma_A^i$, with analogous expressions for cell types B and C, then our model defining $p\left(g_i^{A,B,C} \mid T = \mathcal{A}, \beta_i = 1, \{C\}\right)$ is that either $\mu_B^i < \mu_C^i$ and $\mu_B^i < \mu_A^i$ or $\mu_C^i < \mu_B^i$ and $\mu_C^i < \mu_A^i$, where $\mu_A^i$, $\mu_B^i$ and $\mu_C^i$ are the mean values of the expression levels of $g_i$ in cell types A, B and C. Thus,

$$p\left(g_i^{A,B,C} \mid T = \mathcal{A}, \beta_i = 1, \{C\}\right) = \frac{1}{2} \left\{ \begin{array}{l} p\left(g_i^{A,B,C} \mid \mu_B^i < \mu_A^i, \mu_B^i < \mu_C^i, \{C\}\right) \\ + p\left(g_i^{A,B,C} \mid \mu_C^i < \mu_A^i, \mu_C^i < \mu_B^i, \{C\}\right) \end{array} \right\} \tag{5}$$

The terms in *Equation 5* can be calculated by integrating over the prior probability distribution of the means $\mu_A^i$, $\mu_B^i$ and $\mu_C^i$ and standard deviations $\sigma_A^i$, $\sigma_B^i$ and $\sigma_C^i$, with the conditions on the means constraining the domains of integration:

$$p(g_i^{A,B,C}|T = \mathcal{A}, \beta_i = 1, \{C\})$$
$$= \frac{1}{2} \iiint\limits_{\mu_B^i < \mu_A^i, \mu_B^i < \mu_C^i, \sigma_A^i, \sigma_B^i, \sigma_C^i} D_A(g_i^A|\mu_A^i, \sigma_A^i) D_B(g_i^B|\mu_B^i, \sigma_B^i) D_C(g_i^C|\mu_C^i, \sigma_C^i) p(\mu_A^i, \mu_B^i, \mu_C^i, \sigma_A^i, \sigma_B^i, \sigma_C^i)$$
$$+ \frac{1}{2} \iiint\limits_{\mu_C^i < \mu_A^i, \mu_C^i < \mu_B^i, \sigma_A^i, \sigma_B^i, \sigma_C^i} D_A(g_i^A|\mu_A^i, \sigma_A^i) D_B(g_i^B|\mu_B^i, \sigma_B^i) D_C(g_i^C|\mu_C^i, \sigma_C^i) p(\mu_A^i, \mu_B^i, \mu_C^i, \sigma_A^i, \sigma_B^i, \sigma_C^i) \tag{6}$$

Probabilities $p\left(g_i^{A,B,C} \mid T = \mathcal{B}, \beta_i = 1, \{C\}\right)$ and $p\left(g_i^{A,B,C} \mid T = \mathcal{C}, \beta_i = 1, \{C\}\right)$ are defined similarly.

In addition to topologies $\mathcal{A}$, $\mathcal{B}$ and $\mathcal{C}$, we consider a null hypothesis $\varnothing$ in which transition genes have differential expression levels between states, but these levels are not correlated with any particular topology of states. This corresponds to having gene expression levels from cell-types A, B and C coming from three distributions with no restrictions on the relative order of the three means:

$$p\left(g_i^{A,B,C} \mid T = \varnothing, \beta_i = 1, \{C\}\right)$$
$$= \iiint\limits_{\mu_A^i, \mu_B^i, \ \mu_C^i, \sigma_A^i, \sigma_B^i, \sigma_C^i} D_A(g_i^A \mid \mu_A^i, \sigma_A^i) D_B(g_i^B \mid \mu_B^i, \sigma_B^i) D_C(g_i^C \mid \mu_C^i, \sigma_C^i) p(\mu_A^i, \mu_B^i, \mu_C^i, \sigma_A^i, \sigma_B^i, \sigma_C^i) \tag{7}$$

Note that the probability of the data given the null hypothesis is the average of the probabilities of the data given the non-null hypotheses:

$$p(g_i^{A,B,C}|T = \varnothing, \beta_i = 1, \{C\}) = \frac{1}{3} \sum_{T = A, B, C} p(g_i^{A,B,C}|T, \beta_i = 1, \{C\}) \tag{8}$$

Note that $p(g_i^{A,B,C}|T, \{C\}, \alpha_i = 0, \beta_i = 1)$ depends on both $T$ and $\{C\}$.

## Expression for $p\left(g_i^{A,B,C} \mid T, \{C\}, \alpha_i = 1, \beta_i = 0\right)$ (marker genes)

Our model for marker genes assumes that the probability distribution for the expression level of such genes, $p\left(g_i^{A,B,C} \mid T, \{C\}, \alpha_i = 1, \beta_i = 0\right)$ to be independent of $T$ and to be generated from distributions with two cell-types having a low value and the third a high value (for example, $D_{AB}\left(g_i^{A,B,C} \mid \mu_{AB}^i, \sigma_{AB}^i\right)$ for cell-types A and B and $D_C(g_i^C \mid \mu_C^i, \sigma_c^i)$ for cell-type C, with the constraint $\mu_{AB}^i < \mu_C^i$):

$$p(g_i^{A,B,C}|T, \{C\}, \alpha_i = 1, \beta_i = 0)$$
$$= \frac{1}{3} \iiint\limits_{\mu_{AB}^i < \mu_C^i, \sigma_{AB}^i, \sigma_C^i} D_{AB}(g_i^{AB}|\mu_{AB}^i, \sigma_{AB}^i) D_C(g_i^C|\mu_C^i, \sigma_C^i) p(\mu_{AB}^i, \mu_C^i, \sigma_{AB}^i, \sigma_C^i)$$
$$+ \frac{1}{3} \iiint\limits_{\mu_{AC}^i < \mu_B^i, \sigma_{AC}^i, \sigma_B^i} D_{AC}(g_i^{AC}|\mu_{AC}^i, \sigma_{AC}^i) D_B(g_i^B|\mu_B^i, \sigma_B^i) p(\mu_{AC}^i, \mu_B^i, \sigma_{AC}^i, \sigma_B^i)$$
$$+ \frac{1}{3} \iiint\limits_{\mu_{BC}^i < \mu_A^i, \sigma_{BC}^i, \sigma_A^i} D_{BC}(g_i^{BC}|\mu_{BC}^i, \sigma_{BC}^i) D_B(g_i^A|\mu_A^i, \sigma_A^i) p(\mu_{BC}^i, \mu_A^i, \sigma_{BC}^i, \sigma_A^i) \tag{9}$$

Note that $p\left(g_i^{A,B,C} \mid T, \{C\}, \alpha_i = 1, \beta_i = 0\right) = p\left(g_i^{A,B,C} \mid \{C\}, \alpha_i = 1, \beta_i = 0\right)$ does not depend on $T$ but does depend on $\{C\}$.

## Expression for $p\left(g_i^{A,B,C} \mid T, \{C\}, \alpha_i = 0, \beta_i = 0\right)$ (irrelevant genes)

Our model for genes that are neither marker nor transition genes is that the expression levels of such genes, $p\left(g_i^{A,B,C} \mid T, \{C\}, \alpha_i = 0, \beta_i = 0\right)$, is generated from one single distribution $D_{ABC}\left(g_i^{A,B,C} \mid \mu^i, \sigma^i\right)$:

$$p\left(g_i^{A,B,C} \mid T, \{C\}, \alpha_i = 0, \beta_i = 0\right) = \iint_{\mu^i, \sigma^i} D_{ABC}\left(g_i^{A,B,C} \mid \mu^i, \sigma^i\right) p\left(\mu^i, \sigma^i\right) \tag{10}$$

Note that $p\left(g_i^{A,B,C} \mid T, \{C\}, \alpha_i = 0, \beta_i = 0\right) = p\left(g_i^{A,B,C} \mid \alpha_i = 0, \beta_i = 0\right)$ does not depend on $T$ or $\{C\}$.

## Numerical integration

Each of the probabilities on the right hand side of *Equation 4* is evaluated numerically as above. We assume the distribution of the expression of gene $i$ in cluster $c_A$ $D_A\left(g_i^A \mid \mu_A^i, \sigma_A^i\right)$ to be log-normal. Given $m$ log2-transformed replicate measurements $g_i^A$ of gene expression of gene $i$ in cells belonging to cluster $c_A$, the probability of the data assuming mean $\mu_A^i$ and standard deviation $\sigma_A^i$ is:

$$D_A(g_i^A \mid \mu_A^i, \sigma_A^i) = \left(\frac{1}{\sqrt{2\pi\sigma_A^{i2}}}\right)^m \prod_{g_i^A}^m e^{-\frac{(g_i^A - \mu_A^i)^2}{2\sigma_A^{i2}}}. \tag{11}$$

Distributions $D_B$, $D_C$, $D_{AB}$, $D_{AC}$, $D_{BC}$ and $D_{ABC}$ are defined analogously.

We take the *a priori* probability distribution of $\mu^i$ and $\sigma^i$, $p(\mu^i, \sigma^i)$ as uniform over a certain range of means and standard deviations estimated from the data. For the log2-transformed hematopoietic gene expression data, we take $2 < \mu^i < 14$ and $0 < \sigma^i < 0.75$.

We take the prior probabilities for the distributions in different cell types to be independent: $p(\mu_A^i, \mu_B^i, \mu_C^i, \sigma_A^i, \sigma_B^i, \sigma_C^i) = p(\mu_A^i, \sigma_A^i) \, p(\mu_B^i, \sigma_B^i) \, p(\mu_C^i, \sigma_C^i)$. The constraints on the order of the means are enforced by the domain of integration, and the prior must be properly normalized over this domain. For example, in *Equation 6*,

$$\frac{1}{2} \iiint_{\mu_B^i < \mu_A^i, \mu_B^i < \mu_C^i, \sigma_A^i, \sigma_B^i, \sigma_C^i} p(\mu_A^i, \mu_B^i, \mu_C^i, \sigma_A^i, \sigma_B^i, \sigma_C^i)$$

$$+ \frac{1}{2} \iiint_{\mu_C^i < \mu_A^i, \mu_C^i < \mu_B^i, \sigma_A^i, \sigma_B^i, \sigma_C^i} p(\mu_A^i, \mu_B^i, \mu_C^i, \sigma_A^i, \sigma_B^i, \sigma_C^i) = 1. \tag{12}$$

Integrals were evaluated numerically in MATLAB using trapezoidal integration with step-sizes $\delta\mu$ = 0.05 and $\delta\sigma$ = 0.01.

The choice of a log-normal distribution could potentially confound the results, particularly for single-cell RNA-Seq data with significant zero-inflation. This is a potential area for improvement in our algorithm, but the method can easily be adapted to model different distributions of RNA expression, such as gamma distributions (*Shahrezaei and Swain, 2008*; *Wills et al., 2013*) or beta-Poisson distributions (*Delmans and Hemberg, 2016*; *Vu et al., 2016*). In either case, the probability of the data given different topologies would be computed by numeric integration over the parameters of the distribution, for example, $\alpha$ and $\beta$ for the Gamma distribution, by replacing the log-normal distributions in *Equations 7, 9 and 10* with ones from the appropriate model.

The choice of the right parametric form for single-cell RNA expression is still an area of active research. Our choice of log-normal distributions assumes that higher order moments of the

distributions (beyond standard deviation) ought to have a minimal contribution to the predictions, but we have not tested this extensively.

Although the default prior was the uniform prior, we also implemented an empirical prior $p(\mu, \sigma)$ by estimating the empirical distribution over all the Immgen cell types, using the kernel density estimation code provided in (*Botev et al., 2010*). The resulting hematopoietic lineage tree was identical. Using the kernel-density-estimated empirical prior may provide more stability in future analyses.

## Probability of topology given gene expression and cluster identities $p\left(T \mid \left\{g_i^{A,B,C}\right\}, \{C\}\right)$

We can derive the probability of the topology given the gene expression data and cluster identities $p\left(T \mid \left\{g_i^{A,B,C}\right\}, \{C\}\right)$ by summing over all the $\{\alpha_i\}$ and $\{\beta_i\}$ to find the probability of the data given topology $p\left(\left\{g_i^{A,B,C}\right\} \mid T, \{C\}\right)$:

$$
\begin{aligned}
p\left(\left\{g_i^{A,B,C}\right\} \mid T, \{C\}\right) &= \sum_{\alpha_i, \beta_i} p\left(\left\{g_i^{A,B,C}\right\} \mid T, \{\alpha_i\}, \{\beta_i\}, \{C\}\right) p(\{\alpha_i\}, \{\beta_i\} \mid T, \{C\}) \\
&= \sum_{\beta_1} \sum_{\beta_2} \cdots \sum_{\beta_N} \prod_i p\left(g_i^{A,B,C} \mid T, \alpha_i, \beta_i, \{C\}\right) p(\alpha_i, \beta_i) \\
&= \prod_i \left( \sum_{\beta_i} p\left(g_i^{A,B,C} \mid T, \alpha_i, \beta_i, \{C\}\right) p(\alpha_i, \beta_i) \right)
\end{aligned}
\tag{13}
$$

$$
p\left(\left\{g_i^{A,B,C}\right\} \mid T, \{C\}\right) = \prod_i p\left(g_i^{A,B,C} \mid T, \{C\}\right),
\tag{14}
$$

where the probability of gene expression data for gene $i$ given topology $p\left(g_i^{A,B,C} \mid T, \{C\}\right)$ is obtained by summing $p\left(g_i^{A,B,C} \mid T, \{C\}, \alpha_i, \beta_i\right)$ over $\alpha_i$ and $\beta_i$:

$$
\begin{aligned}
p\left(g_i^{A,B,C} \mid T, \{C\}\right) = \ &p\left(g_i^{A,B,C} \mid T, \{C\}, \alpha_i = 0, \beta_i = 1\right) p(\beta_i = 1) \\
&+ p\left(g_i^{A,B,C} \mid \{C\}, \alpha_i = 1, \beta_i = 0\right) p(\alpha_i = 1, \beta_i = 0) \\
&+ p\left(g_i^{A,B,C} \mid \alpha_i = 0, \beta_i = 0\right) p(\alpha_i = 0, \beta_i = 0).
\end{aligned}
\tag{15}
$$

We consider the odds of a gene being a transition gene $p(\beta_i = 1)/p(\beta_i = 0)$ as a sparsity parameter, which we vary. We take $p(\alpha_i = 0 \mid \beta_i = 0) = p(\alpha_i = 1 \mid \beta_i = 0) = 1/2$.

The probability of topology $T$ given data is proportional to the probability of the data given topology $T$ (using Bayes' rule):

$$
p\left(T \mid \left\{g_i^{A,B,C}\right\}, \{C\}\right) = \frac{p\left(\left\{g_i^{A,B,C}\right\} \mid T, \{C\}\right) p(T)}{p\left(\left\{g_i^{A,B,C}\right\} \mid \{C\}\right)},
\tag{16}
$$

where

$$
p\left(\left\{g_i^{A,B,C}\right\} \mid \{C\}\right) = \sum_T p(T) p\left(\left\{g_i^{A,B,C}\right\} \mid T, \{C\}\right).
\tag{17}
$$

Therefore, using *Equation 14*, we obtain the following expression for $p\left(T \mid \left\{g_i^{A,B,C}\right\}, \{C\}\right)$:

$$
p\left(T \mid \left\{g_i^{A,B,C}\right\}, \{C\}\right) = \frac{p(T) \prod_i p\left(g_i^{A,B,C} \mid T, \{C\}\right)}{\sum_T p(T) \prod_i p\left(g_i^{A,B,C} \mid T, \{C\}\right)}.
\tag{18}
$$

*Equation 18* can be written more explicitly by rewriting *Equation 15* as follows:

$$
\begin{aligned}
p\left(g_i^{A,B,C} \mid T, \{C\}\right) &= p\left(g_i^{A,B,C} \mid T, \beta_i = 0, \{C\}\right) p(\beta_i = 0) + p\left(g_i^{A,B,C} \mid T, \beta_i = 1, \{C\}\right) p(\beta_i = 1) \\
&= p\left(g_i^{A,B,C} \mid \beta_i = 0, \{C\}\right) p(\beta_i = 0) \left(1 + \frac{p\left(g_i^{A,B,C} \mid T, \beta_i=1, \{C\}\right) p(\beta_i=1)}{p\left(g_i^{A,B,C} \mid \beta_i=0, \{C\}\right) p(\beta_i=0)}\right).
\end{aligned}
\tag{19}
$$

Here we have used the fact noted earlier that in our generating model, $p\left(g_i^{A,B,C} \mid T, \alpha_i = 1, \ \beta_i = 0, \ \{C\}\right) = p\left(g_i^{A,B,C} \mid \alpha_i = 1, \ \beta_i = 0, \{C\}\right)$ and $p\left(g_i^{A,B,C} \mid T, \alpha_i = 0, \ \beta_i = 0, \ \{C\}\right) = p\left(g_i^{A,B,C} \mid \alpha_i = 0, \ \beta_i = 0\right)$ do not depend on $T$ (*Equation 10*). The terms $\prod_i p\left(g_i^{A,B,C} \mid \beta_i = 0, \{C\}\right) p(\beta_i = 0)$ cancel out in the numerator and denominator of *Equation 18*, and we can write *Equation 18* in terms of ratios of the probabilities of the data given transition-gene and non-transition-gene status:

$$
p\left(T \mid \left\{g_i^{A,B,C}\right\}, \{C\}\right) = \frac{p(T) \prod_i \left(1 + \frac{p\left(g_i^{A,B,C} \mid T, \beta_i=1, \{C\}\right) p(\beta_i=1)}{p\left(g_i^{A,B,C} \mid \beta_i=0, \{C\}\right) p(\beta_i=0)}\right)}{\sum_T p(T) \prod_i \left(1 + \frac{p\left(g_i^{A,B,C} \mid T, \beta_i=1, \{C\}\right) p(\beta_i=1)}{p\left(g_i^{A,B,C} \mid \beta_i=0, \{C\}\right) p(\beta_i=0)}\right)}.
\tag{20}
$$

We can rewrite *Equation 20* as:

$$
p\left(T \mid \left\{g_i^{A,B,C}\right\}, \{C\}\right) = \frac{p(T) \prod_i \left(1 + \frac{1}{p(T)} \mathcal{O}_i \, p\left(T \mid g_i^{A,B,C}, \beta_i = 1, \{C\}\right)\right)}{\sum_T p(T) \prod_i \left(1 + \frac{1}{p(T)} \mathcal{O}_i \, p\left(T \mid g_i^{A,B,C}, \beta_i = 1, \{C\}\right)\right)},
\tag{21}
$$

where $\mathcal{O}_i$ is the odds that gene $i$ is a transition gene, given clustering:

$$
\mathcal{O}_i = \frac{p\left(\beta_i = 1 \mid g_i^{A,B,C}, \{C\}\right)}{p\left(\beta_i = 0 \mid g_i^{A,B,C}, \{C\}\right)} = \frac{p\left(g_i^{A,B,C} \mid \beta_i = 1, \{C\}\right)}{p\left(g_i^{A,B,C} \mid \beta_i = 0, \{C\}\right)} \frac{p(\beta_i = 1)}{p(\beta_i = 0)}
\tag{22}
$$

and $p\left(T \mid g_i^{A,B,C}, \beta_i = 1, \{C\}\right)$ is the probability of $T$ given only gene expression data for gene $i$, clustering and that gene $i$ is a transition gene:

$$
p\left(T \mid g_i^{A,B,C}, \beta_i = 1, \{C\}\right) = \frac{p\left(g_i^{A,B,C} \mid \beta_i = 1, \ T, \{C\}\right) p(T)}{p\left(g_i^{A,B,C} \mid \beta_i = 1, \{C\}\right)}.
\tag{23}
$$

Thus each gene's contribution $p\left(T \mid g_i^{A,B,C}, \beta_i = 1, \{C\}\right)$ to the probability of the topology given total gene expression $p\left(T \mid \left\{g_i^{A,B,C}\right\}, \{C\}\right)$ is weighted by the odds $\mathcal{O}_i$ that it is transition gene.

### Rewriting *Equation 21* in terms of negative votes

Let us denote the probability of gene expression data for gene $i$ given that cell cluster $\xi$ has the distribution with minimum mean expression as $p\left(g_i^{A,B,C} \mid \mu_\xi^i \text{ is min}, \{C\}\right)$. For example, $p\left(g_i^{A,B,C} \mid \mu_B^i \text{ is min}, \{C\}\right) = p\left(g_i^{A,B,C} \mid \mu_B^i < \mu_A^i, \mu_B^i < \mu_C^i, \{C\}\right)$. Then, using $p(T) = 1/4$ and *Equations 5 and 8*, we can write:

$$p\left(g_i^{A,B,C}\mid\beta_i=1,\{C\}\right)\;=\frac{1}{4}\left[\sum_{T=\mathcal{A},\mathcal{B},\mathcal{C},\varnothing}p\left(g_i^{A,B,C}\mid T,\beta_i=0,\{C\}\right)\right]$$

$$=\frac{1}{4}\left[\underbrace{\sum_{\xi=A,B,C}p\left(g_i^{A,B,C}\mid\mu_\xi^i\text{ is min, }\{C\}\right)}_{T=\mathcal{A},\mathcal{B},\mathcal{C}}+\frac{1}{3}\underbrace{\sum_{\xi=A,B,C}p\left(g_i^{A,B,C}\mid\mu_\xi^i\text{ is min, }\{C\}\right)}_{T=\varnothing}\right] \quad (24)$$

$$=\frac{1}{3}\left[\sum_{\xi=A,B,C}p\left(g_i^{A,B,C}\mid\mu_\xi^i\text{ is min, }\{C\}\right)\right]$$

Therefore, for $T=\mathcal{A},\mathcal{B},\mathcal{C}$, we can rewrite *Equation 5* as:

$$p\left(g_i^{A,B,C}\mid T,\beta_i=1,\{C\}\right)=\frac{1}{2}\left[3\,p\left(g_i^{A,B,C}\mid\beta_i=1,\{C\}\right)-p\left(g_i^{A,B,C}\mid\mu_T^i\text{ is min, }\{C\}\right)\right] \quad (25)$$

Combining *Equations 21 and 25*, we derive, for $T\neq\varnothing$:

$$p\left(T\mid\{g_i^{A,B,C}\},\{C\}\right)\;\propto p(T)\prod_i\left(1+\frac{p\left(g_i^{A,B,C}\mid T,\beta_i=1,\{C\}\right)}{p\left(g_i^{A,B,C}\mid\beta_i=1,\{C\}\right)}\mathcal{O}_i\right)$$

$$\propto p(T)\prod_i\left(1+\frac{\frac{1}{2}\left[3\,p\left(g_i^{A,B,C}\mid\beta_i=1,\{C\}\right)-p\left(g_i^{A,B,C}\mid\mu_T^i\text{ is min, }\{C\}\right)\right]}{p\left(g_i^{A,B,C}\mid\beta_i=1,\{C\}\right)}\mathcal{O}_i\right) \quad (26)$$

$$\propto p(T)\prod_i\left(1+\tfrac{3}{2}\mathcal{O}_i\left[1-p\left(\mu_T^i\text{ is min}\mid g_i^{A,B,C},\beta_i=1,\{C\}\right)\right]\right),$$

where $p\left(\mu_T^i\text{ is min}\mid g_i^{A,B,C},\beta_i=1,\{C\}\right)$ is the probability that cell cluster $T$ (the intermediate cluster in topology $T$) has the distribution with the minimum mean for gene $i$:

$$p\left(\mu_T^i\text{ is min}\mid g_i^{A,B,C},\beta_i=1,\{C\}\right)\;=\frac{p\left(g_i^{A,B,C}\mid\mu_T^i\text{ is min},\{C\}\right)p\left(\mu_T^i\text{ is min}\mid\beta_i=1,\{C\}\right)}{p\left(g_i^{A,B,C}\mid\beta_i=1\right)}$$

$$=\frac{1}{3}\frac{p\left(g_i^{A,B,C}\mid\mu_T^i\text{ is min},\{C\}\right)}{p\left(g_i^{A,B,C}\mid\beta_i=1,\{C\}\right)}. \quad (27)$$

Every gene can be thought of as casting a vote $-p\left(\mu_T^i\text{ is min}\mid g_i^{A,B,C},\beta_i=1,\{C\}\right)$ against cell type $T$ being the intermediate, and this vote is weighted by the odds $\mathcal{O}_i$ of the gene $i$ being a transition gene and having a unique minimum, given the clustering. This corresponds to *Equation 1* in the main text.

## Expression for $p\left(T,\{\alpha_i\},\{\beta_i\}\mid\{g_i^{A,B,C}\},\{C\}\right)$

Once $p\left(T\mid\{g_i^{A,B,C}\},\{C\}\right)$ is calculated, it is straightforward to find $p\left(T,\{\alpha_i\},\{\beta_i\}\mid\{g_i^{A,B,C}\},\{C\}\right)$:

$$p\left(T,\{\alpha_i\},\{\beta_i\}\mid\{g_i^{A,B,C}\},\{C\}\right)=p\left(\{\alpha_i\},\{\beta_i\}\mid\{g_i^{A,B,C}\},T,\{C\}\right)p\left(T\mid\{g_i^{A,B,C}\},\{C\}\right), \quad (28)$$

where $p\left(\{\alpha_i\},\{\beta_i\}\mid\{g_i^{A,B,C}\},T,\{C\}\right)$ is the probability of $\{\alpha_i\}$ and $\{\beta_i\}$ given the particular topology $T$, clustering $\{C\}$ and gene expression. Because we have assumed that gene expression patterns $p\left(g_i^{A,B,C}\mid T,\{C\},\alpha_i,\beta_i\right)$ are conditionally independent given $T$, $\{C\}$, $\alpha_i$ and $\beta_i$ (*Equation 3*), the probabilities of being marker or transition genes $\alpha_i$ or $\beta_i$ are also conditionally independent given gene expression, clustering and the topology:

$$p\left(\{\alpha_i\},\{\beta_i\}\mid\{g_i^{A,B,C}\},T,\{C\}\right)\;=\frac{p\left(\{g_i^{A,B,C}\}\mid T,\{\alpha_i\},\{\beta_i\},\{C\}\right)p\left(\{\alpha_i\},\{\beta_i\}\mid T,\{C\}\right)}{p\left(\{g_i^{A,B,C}\}\mid T,\{C\}\right)}$$

$$=\prod_i\frac{p\left(g_i^{A,B,C}\mid T,\alpha_i,\beta_i,\{C\}\right)p\left(\alpha_i,\beta_i\mid T,\{C\}\right)}{p\left(g_i^{A,B,C}\mid T,\{C\}\right)} \quad (29)$$

$$=\prod_i p\left(\alpha_i,\beta_i\mid T,\{C\},g_i^{A,B,C}\right),$$

where $p\left(\alpha_i, \beta_i \mid T, \{C\}, g_i^{A,B,C}\right)$ is the probability that gene $i$ is a marker or transition gene given its gene expression, the clustering, and that the topology is $T$.

## Choice of prior odds does not affect the most likely topology

The only free parameter in our calculation above is the prior odds of gene $i$ being a transition gene, $p(\beta_i = 1)/p(\beta_i = 0)$. At one extreme, if $p(\beta_i = 1)/p(\beta_i = 0) \to 0$, then $p\left(T \mid \left\{g_i^{A,B,C}\right\}\right) \to p(T)$: if we assume that none of the genes are transition genes, then knowing gene expression does not give us any new knowledge of the topology $T$, since only transition genes are informative about $T$. At the other extreme, if $p(\beta_i = 1)/p(\beta_i = 0) \to \infty$ then the null hypothesis dominates: if all genes are transition genes, then there will be negative votes against all topologies. We computed the behavior of $p\left(T \mid \left\{g_i^{A,B,C}\right\}\right)$ between these two limits to determine the sensitivity of our answer to $p(\beta_i = 1)/p(\beta_i = 0)$.

*Figure 2—figure supplement 1* shows the dependence of the probabilities $p\left(T \mid \left\{g_i^{A,B,C}\right\}\right)$ on the prior odds for triplets CMP/ST/MPP and GMP/MEP/FrBC for values of $p(\beta_i = 1)/p(\beta_i = 0)$ between $10^{-8}$ and $10^2$. For triplet CMP/ST/MPP the topology $ST \equiv CMP - ST - MPP$ dominates for $p(\beta_i = 1)/p(\beta_i = 0)$ between $10^{-2}$ and 10, whereas for triplet MEP/GMP/FrBC there is no value of the prior odds that strongly favors a non-null topology. For most triplets, the most likely topology does not depend on the choice of prior odds; when building lineage trees, we ignore those triplets where different choices of prior odds lead to different most-likely topologies, i.e. there is more than one non-null topology that reaches probability 0.6 over the range of prior odds.

## Determination of lineage tree from triplet topologies

### Selection of triplets

In order to build lineage trees from the topologies we determine for each cell type, we select the triplets for which our determination of the topology is most robust. There is one free parameter in our model: the prior odds for a gene to be a transition gene in the absence of gene expression data, $p(\beta_i = 1)/p(\beta_i = 0)$. For each triplet, we vary this parameter between $10^{-6}$ and $10^2$ and calculate the probability of the topology given gene expression data $p\left(T \mid \left\{g_i^{A,B,C}\right\}\right)$ as a function of the prior odds.

We want to consider only triplets which showed a single dominant topology. We exclude triplets which show a weak probability for a particular topology or ones which depend on a particular choice of prior odds. We also do not consider triplets which show a strong probability for two different topologies, depending on the choice of prior odds.

There were 14 such triplets among the 165 hematopoietic triplets. Mathematically, these cases come up when the genes that are most likely to show the clear minimum pattern (furthest on the right in a 'dot plot') suggest one topology, but if one used a more permissive value of the sparsity parameter, a different topology wins out. One of the cell types might have a small number of genes with very high odds, but then fewer genes with moderately high odds compared to the other cell types. We did not notice a clear pattern in the identity of the triplets exhibiting this behavior, but 9 of the 14 were triplets of length five or greater in the Adolfsson model. One of the triplets with this behavior was the MLP/CMP/GMP triplet, and the dominant topology was either MLP (at low prior odds) or CMP (at higher prior odds). Interestingly, both cell types are progenitors to GMP in the Adolfsson model.

We consider triplets for which only one non-null topology has probability $p\left(T \mid \left\{g_i^{A,B,C}\right\}\right)$ greater than 0.6. The probabilities of the different topologies for each triplet in the hematopoietic tree are shown in *Figure 2—source data 1* and for the triplets in the cortical development tree in *Figure 4—source data 2*.

### Pruning rule

We assemble the triplets with known topology into an undirected graph. Since we determined topologies by considering cell types three at a time, we obtain topological relationships involving both cell types that are nearest neighbors and cell types that are more distantly related. In order to

reconstruct the tree, we must determine which cell types are nearest neighbors and which ones are separated by one or more intermediate cell types.

The set of inferred topologies allows us to determine which cell types are separated by intermediates. For every pair of cell types, we ask whether any of the inferred topologies features an intermediate between the two cell types. If such a topology has been inferred, we consider that the two cell types are not nearest neighbors, and that at least one other cell type is an intermediate. For example, we can ignore triplet CMP – **LT** – MPP because triplet LT – **ST** – CMP testifies that there exists an intermediate between LT and CMP, and triplet LT – **ST** – MPP testifies that there exists an intermediate between LT and MPP (*Figure 3—figure supplement 1*).

Note that this pruning rule does not assume the absence of loops. The lineage tree we infer for the hematopoietic progenitors contains a loop that includes ST to CMP to GMP on one side and ST to MPP to MLP to GMP on the other side. The loop involves triplets CMP – **ST** – MPP, ST – **CMP** – GMP, ST – **MPP** – MLP and MPP – **MLP** – GMP (we cannot determine the topology of triplet CMP/MLP/GMP). None of the triplets shows a topology that would allow us to break up the loop.

## Distinguishing between two models of hematopoiesis

The topologies we infer support the model from *Adolfsson et al. (2005)*, in which CMP splits from ST-HSC. In particular, there are several triplets that can distinguish the Adolfsson model from the traditional picture, and they support the Adolfsson model. These triplets include CMP – **ST** – MPP, CMP – **ST** – MLP, MEP – **ST** – MLP and CMP – **LT** – MPP. These triplets show that, unlike in the traditional picture, cell types CMP and MEP split from ST are not descended from MPP or MLP.

On the other hand, triplets LT – **MLP** – GMP, ST – **MLP** – GMP and MPP – **MLP** – GMP show that MLP is an intermediate between the earliest progenitors and GMP. See also *Figure 2—figure supplement 2* and *Figure 2—source data 2*.

## Stability analysis

The inference algorithm depends on several parameters and priors. We performed a stability analysis for both the microarray hematopoietic data and the human brain single cell data to determine the parameter ranges for which the inferred lineage tree was unchanged.

1. Hematopoiesis
   - For the prior probability, given that a gene is not a transition gene, that it is an irrelevant gene, our default value was $p(\alpha_i = 0 \mid \beta_i = 0) = 0.5$, but the tree was unchanged for values of $p(\alpha_i = 0 \mid \beta_i = 0)$ between 0.25 and 0.65.
   - For the prior probabilities of different topologies, our default value was $p(\emptyset) = p(\mathcal{A}) = p(\mathcal{B}) = p(\mathcal{C}) = 0.25$. We varied $p(\emptyset)$ while keeping the prior probabilities of the non-null topologies equal: $p(T \neq \emptyset) = \frac{1}{3}(1 - p(\emptyset))$. The tree was unchanged for values of $p(\emptyset)$ between 0.1 and 0.35.
   - We used a threshold of 0.6 to consider a triplet significant for the tree-building step. The tree was unchanged for thresholds between 0.5 and 0.65.
   - A key input parameter into the algorithm is the expected prior distribution of means and standard deviations $p(\mu, \sigma)$ used in the numeric integration. Our default prior was a uniform prior for both means and standard deviations over reasonable ranges of these parameters. We also implemented an empirical prior $p(\mu, \sigma)$ by estimating the empirical distribution over all the Immgen cell types, using kernel density estimation (*Botev et al., 2010*). The resulting hematopoietic lineage tree was identical. Using the kernel-density-estimated empirical prior may provide more stability in future analyses.
2. Brain Development
   - For the prior probability, given that a gene is not a transition gene, that it is an irrelevant gene, our default value was $p(\alpha_i = 0 \mid \beta_i = 0) = 0.5$, but the tree was unchanged for values of $p(\alpha_i = 0 \mid \beta_i = 0)$ between 0.02 and 0.97.
   - For the prior probabilities of different topologies, our default value was $p(\emptyset) = p(\mathcal{A}) = p(\mathcal{B}) = p(\mathcal{C}) = 0.25$. We varied $p(\emptyset)$ while keeping the prior probabilities of the non-null topologies equal: $p(T \neq \emptyset) = \frac{1}{3}(1 - p(\emptyset))$. The tree was unchanged for values of $p(\emptyset)$ between 0.06 and 0.96.
   - We used a threshold of 0.6 to consider a triplet significant for the tree-building step. The tree was unchanged for thresholds between 0.5 and 0.8.

- For the single cell data, we also changed the number of initial seed clusters for the iterative algorithm within a range from 30 to 45. In each case, the tree remained the same, and never more than 23% of the cells were clustered into a different cell type.

## Acknowledgements

We thank Sandeep Choubey, Alex Schier, Andrew Murray, David Scadden and Toshihiko Oki for scientific discussions. We also thank Christof Koch, Andrew Murray, KC Huang, Jim Valcourt and Dann Huh for detailed comments and feedback on this work. We are grateful to Simona Lodato for helping us interpret our results and in particular help us write the section on the genes we discover to be important during cortical development. We are particularly grateful to the Senior and Reviewing Editors and to the four peer reviewers, including Nir Yosef, for their very thoughtful comments and suggestions. The study was supported by NSF-GRFP and NDSEG Fellowships (LF), an NIH Pioneer Award (SR), and the Allen Institute for Brain Science (SR).

## Additional information

### Funding

| Funder | Author |
| --- | --- |
| National Science Foundation | Leon A. Furchtgott |
| National Institutes of Health | Sharad Ramanathan |
| Allen Foundation | Vilas Menon<br>Sharad Ramanathan |

The funders had no role in study design, data collection and interpretation, or the decision to submit the work for publication.

### Author contributions

LAF, Conceptualization, Data curation, Software, Formal analysis, Validation, Investigation, Visualization, Methodology, Writing—original draft, Writing—review and editing; SM, Conceptualization, Data curation, Software, Formal analysis, Validation, Visualization, Methodology, Writing—original draft, Writing—review and editing; VM, Validation, Visualization, Writing—review and editing; SR, Conceptualization, Resources, Formal analysis, Supervision, Funding acquisition, Investigation, Visualization, Methodology, Writing—original draft, Project administration, Writing—review and editing

### Author ORCIDs

Leon A Furchtgott, http://orcid.org/0000-0002-4258-0950

## Additional files

### Major datasets

The following previously published datasets were used:

| Author(s) | Year | Dataset title | Dataset URL | Database, license, and accessibility information |
| --- | --- | --- | --- | --- |
| Gopalan G | 2009 | Immunological Genome Project | http://www.ncbi.nlm.nih.gov/geo/query/acc.cgi?acc=GSE15907 | Publicly available at the NCBI Gene Expression Omnibus (accession no: GSE15907) |
| Grün D, Lyubimova A, Kester L, Wiebrands K, Basak O, Sasaki N, Clevers H, van Oudenaarden A | 2015 | Single-Cell mRNA Sequencing Reveals Rare Intestinal Cell Types | https://www.ncbi.nlm.nih.gov/geo/query/acc.cgi?acc=GSE62270 | Publicly available at the NCBI Gene Expression Omnibus (accession no: GSE62270) |
| Yao Z, Mich JK, Ku | 2016 | Region-specific neural stem cell | https://www.ncbi.nlm. | Publicly available at |

S, Menon V, Krostag A, Martinez RA, Grimley JS, Wang Y, Ramanathan S, Levi BP | lineages revealed by single-cell rna-seq from human embryonic stem cells [Smart-seq] | nih.gov/geo/query/acc.cgi?acc=GSE86982 | the NCBI Gene Expression Omnibus (accession no: GSE86982)

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
