## [Decision Letter]

Thank you for submitting your article "Discovering sparse transcription factor codes for cell states and state transitions during development" for consideration by *eLife*. Your article has been favorably evaluated by Arup Chakraborty (Senior Editor) and three reviewers, one of whom, Nir Yosef (Reviewer #1), is a member of our Board of Reviewing Editors.

The reviewers have discussed the reviews with one another and the Reviewing Editor has drafted this decision to help you prepare a revised submission.

Summary:

This paper presents an interesting framework for identification of lineage/ developmental relationships based on whole transcriptome profiles. The basic notion is that differentially expressed genes among triplets of samples (representing either a developmental fork or a linear developmental progression) should largely exhibit a prototypical pattern that reflects the lineage relationship. The authors successfully apply this to infer lineage relationship in hematopoiesis, using bulk- level RNA-seq. They then turn to apply their method to single cell data of early neuronal development, and concomitantly infer clusters of cells (each representing a developmental state), and their organization into a lineage structure. A subset of these results is followed up in another, yet unpublished manuscript. The algorithm was also used and followed up on in a companion manuscript of ESC differentiation.

Essential revisions:

Overall, this study is innovative and provides new insights into cell-type detection and lineage relationship. The method developed here is complementary to existing pseudotime inference methods in single-cell analysis and the paper is clearly written. Nonetheless, the paper may benefit from additional valuation of the method's performance, specifically – by applications to additional data sets and by systematic comparison with existing methods. More detailed descriptions of their methods is also merited, as described below.

1) "Lineage progression" triplets: In the case of cell type B that gives rise to cell type A that gives rise to cell type C the middle node (A) is chosen as the "root"

1.1) The terminology "root cell type" and "terminal leaf cell types" seems inappropriate and confusing in this case and should be revised.

1.2) The minimum-in-leaf model makes sense in the "Lineage progression" case as it captures the common scenario of genes that get turned on or off in a lineage path, with some persistence of this state. However, in such triplets that are distant enough along the path this may contradict the single "pulse" model (i.e., transient change) proposed in other studies. To explore this, Figure 1 should be revised. Currently it only shows the majority statistics. It would be important to see the fraction of genes that conform to the minimum-in-leaf model (which can be anywhere between 51% and 100%) while contrasting:

1.2.1) "Lineage progression" vs. "cell fate decision" triplets.

1.2.2) Triplets with leaves vs. triplets that only include internal nodes.

1.2.3) Triplets with different distance in the tree.

2) The Clear-minimum-in-leaf model.

2.1) To further support the validity of the minimum-in-leaf model, the latter extension to Figure 1 (comment 1.2) should also include comparison between all related triplets vs. all unrelated triplets (i.e., representatives from three different branches).

2.2) Subsection “Discovering sparse patterns correlated with lineage transitions”, third paragraph: There are more than 2 possible patterns for differentially expressed genes between 3 cell types. In addition to the clear minimum A<B=C and clear maximum A=B<C there is also A<B<C, which is actually what is expected for lateral inhibition, discussed in the text several times. This pattern of three clearly distinct expression levels is not discussed at all in the text (in fact, it is excluded from the model, as described in subsection “1. Notation; Bayes’ Rule”, first paragraph). A clear example is HLF, mentioned in the text as higher in MPP compared to CMP, but it is much higher in the third member of the discussed triplet, ST (according to ImmGen skyline browser).

2.3) The B&T subtree used for the observation has a very simple topology of only two branches. That might limit the generality of the conclusion. Indeed, the reconstructed lineage tree of the progenitors include a loop, which is not supported by the literature. Authors should discuss this difference from known tree topology. Also, why did they use their approach to reconstruct only hematopoietic progenitors and not all ~250 ImmGen populations?

3) First algorithm.

3.1) For the formula in the third paragraph of the subsection “Using patterns to infer lineages”, but I wonder how to draw a cutoff for a tree to be 'correct'. If the method is applied to unrelated cell-types, what is distribution of the p-values?

3.2) How robust is the lineage prediction model to the initialization procedure and parameter choices? For example, how sensitive is the resulting tree on the number of initial clusters? how sensitive are the results for the algorithm's parameters (e.g., probability cutoffs) or for sub-sampling of samples or genes? A similar stability analysis should also be included for the second algorithm (extension to single cells), which can be even more sensitive due to its iterative nature.

4) Application to hematopoiesis:

4.1) It is not clear how well the algorithm can separate related from unrelated triplets? (i.e., representatives from three different branches). While this is mentioned anecdotally in Figure 2, it should be evaluated systematically, e.g., with AUROC.

4.2) Given the constraints of the algorithm, the aggregated lineage tree is undirected. While this is mentioned briefly, this key point should be better clarified in the main text.

5) Algorithm's extension to single cells.

5.1) Comparison to other methods.

Recently a number of papers have been published to infer cell lineages from single-cell gene expression data, such as Monocole (Trapnell et al. NBT, 2014), Treutlein et al. (Nature 2014), Wishbone (Setty, Nature Biotech, 2016), tSCAN (Ji, NAR, 2016), StemID (Grun, Cell Stem Cell 2016), and diffusion map (Haghverdi, Nature Methods, 2016). It will be useful to compare with these methods at least to get a sense to what degree this method provides new information (note: it looks like Figure 4 comes from Monocole, but we could not find a mention for it in the Results section).

5.2) Differential expression in single cells.

The t-test used to identify clear minimum and clear maximum patterns is appropriate for microarray analysis, but one limitation of t-test is it assumes the underlying distribution is normal, which is not true for RNAseq, and especially single-cell RNA-seq data. Specifically, single-cell RNA-seq can be severely under-powered because many genes (especially transcription factors) have low or even zero expression values due to dropout. Please discuss ways to incorporate more sophisticated tests that addresses this challenge (e.g., as in scDE), or demonstrate why this is not needed.

5.3) Related to that – the threshold used for definition of a pattern is p<0.005 in a two sample t-test. Though not specified by the authors, it seems that as there are 1500 transcription factors tested, and 3 tests performed per gene (to test the separation of each of the 3 triplet members from the other two), 4500 tests were performed per triplet. Thus, one would expect to find 22 transcription factors (4500*0.005) with a pattern that pass this threshold for each triplet in the absence of a signal. The authors have to address the multiple comparisons issue.

6) Application on Single cell data

6.1) While the single-cell data comes from another manuscript that is under review elsewhere; the information provided on this data is insufficient.

6.1.1) Please provide a compete "census" of the data – how many cells from each condition; how many reads per library, which read lengths were used?

6.1.2) How was the data processed? Should we worry about potential confounders of library quality (which have been presented and discussed time and again in the recent scRNA-seq literature)?

6.2) The algorithm should have been demonstrated first on a single cell RNA-seq datasets where the true cell type transitions tree is known, such as one of the publicly available hematopoietic progenitors single cell RNA-seq datasets.

6.3) There is no estimate of how sparse is the resulting code – how many genes can be used to decipher the topology? Can they be identified ahead? Is it possible that any group of genes of similar size (~1500) can code for cell states and cell transitions with the same success as the transcription factors?

6.4) It is not clear how the tree was generated. Was it using the generic algorithm from the previous section, or by enforcing the temporal order of the samples? If it is the latter – please describe the procedure and explain why the generic algorithm does not work.

---

## [Author Response]

*Essential revisions:*

*Overall, this study is innovative and provides new insights into cell-type detection and lineage relationship. The method developed here is complementary to existing pseudotime inference methods in single-cell analysis and the paper is clearly written. Nonetheless, the paper may benefit from additional valuation of the method's performance, specifically – by applications to additional data sets and by systematic comparison with existing methods. More detailed descriptions of their methods is also merited, as described below.*

We would like to thank the Senior and Reviewing Editors and four peer reviewers for their thoughtful comments and suggestions. The feedback and suggestions have greatly improved our manuscript as well as strengthened our conclusions and for this again we are grateful. We have considered each comment and have amended the manuscript to add 5 new figure supplements, 3 new source data items, and 1 movie, as well as edited the text as noted in the detailed responses below.

*1) "Lineage progression" triplets: In the case of cell type B that gives rise to cell type A that gives rise to cell type C the middle node (A) is chosen as the "root"*

*1.1) The terminology "root cell type" and "terminal leaf cell types" seems inappropriate and confusing in this case and should be revised.*

We would be happy to change the terminology to something clearer, but we have struggled to find an unequivocally clearer alternative. We would like to propose a few alternatives, which we feel all have advantages and disadvantages, and we hope the reviewers could advise on which conveys the meaning most accurately (or propose a better alternative): A) central node and peripheral nodes. B) intermediate node and endpoints. C) internal node and leaf nodes.

D) central node and terminal nodes.

We have also clarified our use of the terms “root” and “leaf, adding: “Note that even in the case in which cell type B gives rise to cell type A which gives rise to cell type C, we will refer to A as the “root” and B and C as “leaves” for that particular triplet.”

1.2) The minimum-in-leaf model makes sense in the "Lineage progression" case as it captures the common scenario of genes that get turned on or off in a lineage path, with some persistence of this state. However, in such triplets that are distant enough along the path this may contradict the single "pulse" model (i.e., transient change) proposed in other studies. To explore this, Figure 1 should be revised. Currently it only shows the majority statistics. It would be important to see the fraction of genes that conform to the minimum-in-leaf model (which can be anywhere between 51% and 100%) while contrasting:

*1.2.1) "Lineage progression" vs. "cell fate decision" triplets.*

*1.2.2) Triplets with leaves vs. triplets that only include internal nodes.*

*1.2.3) Triplets with different distance in the tree.*

We added an additional figure supplement to Figure 1 (Figure 1—figure supplement 2) to test these questions. Figure 1—figure supplement 2 shows the number of genes conforming to the pattern for each triplet, and Figure 1—figure supplement 2 show the triplets shaded according to decision type, presence of internal nodes, and distance. Although biologically, each of these subsets could reasonably be expected to exhibit different statistics, we found that each scenario produced aggregate statistics that were indistinguishable from Figure 1. Since Figure 1—figure supplement 2 includes plots for each of the scenarios suggested in this comment, we have left the representation in Figure 1 unchanged for the purposes of clarity. [Supplementary-material SD1-data] has been appropriately updated.

*2) The Clear-minimum-in-leaf model.*

*2.1) To further support the validity of the minimum-in-leaf model, the latter extension to Figure 1 (comment 1.2) should also include comparison between all related triplets vs. all unrelated triplets (i.e., representatives from three different branches).*

The comment suggests we compare the model’s success on triplets with known relationships vs. those with unknown relationships to establish an expectation of the false positives from our algorithm. This is an important point that we have addressed in two ways.

First, we have repeated the analysis in Figure 1 on a set of 100 unrelated triplets from the same hematopoietic lineage tree, where there is no clear relationship between the cell types. [Supplementary-material SD1-data] has been updated with these triplets. We found that while there were a substantial number of genes exhibiting a clear minimum in one of the three cell types in unrelated triplets, their minima were evenly distributed amongst the cell types (in contrast to what is seen in related triplets: the clear minimum pattern is not seen in the ‘root’). To quantify this, we counted the fraction of genes *f_i_*which reached a clear minimum in cell type *i = A, B, C* (For A, B, and C unrelated cell types), and for each triplet, we computed the entropy 𝑆 = −∑_i=_*_A,B,C_ f_i_* log (*f_i_*) and plotted this quantity vs. the number of genes showing a minimum in any triplet for unrelated and related triplets (Figure 1—figure supplement 3). The unrelated triplets have higher entropy and typically more genes showing the downregulation pattern.

Second, we applied our Bayesian methodology to the set of 150 related and 100 unrelated triplets and compared the distribution of the probability of obtaining a non-null topology. (Figure 1—figure supplement 3). The two distributions diverge considerably, with an AUROC (Area Under Receiver Operating Characteristic) of 0.96, which is shown in a new supplemental figure (Figure 1—figure supplement 3).

*2.2) Subsection “Discovering sparse patterns correlated with lineage transitions”, third paragraph: There are more than 2 possible patterns for differentially expressed genes between 3 cell types. In addition to the clear minimum A<B=C and clear maximum A=B<C there is also A<B<C, which is actually what is expected for lateral inhibition, discussed in the text several times. This pattern of three clearly distinct expression levels is not discussed at all in the text (in fact, it is excluded from the model, as described in subsection “1. Notation; Bayes’ Rule”, first paragraph). A clear example is HLF, mentioned in the text as higher in MPP compared to CMP, but it is much higher in the third member of the discussed triplet, ST (according to ImmGen skyline browser).*

This is an important comment and we have tried to clarify the text accordingly. First, we are defining the clear minimum pattern as A < B & A < C and the clear maximum pattern as A < C & B < C. (The p-value we calculate for the clear minimum model is the less significant of the two p-values for ttest(A,B) and ttest(A,C)). Given this definition, a gene displaying the A < B < C pattern would fulfill both the clear minimum and clear maximum patterns.

In our inference method, transition genes are similarly defined as having a minimum expression level, i.e. A < B and A < C. These constraints are enforced in the bounds of the numeric integration (point 3 of the subsection “Bayesian Framework for inferring cluster identities, state transitions, and marker and transition genes simultaneously”), and we are not constraining B=C. Indeed, Hlf is identified with high probability as a transition gene for triplet ST/MPP/CMP despite the two levels in ST and MPP. We have revised the text to discuss and clarify genes with three distinct levels (subsection “Discovering sparse patterns correlated with lineage transitions”, third paragraph).

In response to the comments we defined the A < B < C pattern as the “clear median” pattern, and we have evaluated how well the pattern correlates with the known topologies. Specifically, does the cell type with median expression level tend to be the root cell type of the triplet? We found that, similar to the clear maximum pattern, the clear median pattern was not a good predictor of the topology.

Therefore, for clarity, we did not define an additional clear median pattern in the main text, and clarified that genes with the clear median pattern belonged to both the clear minimum and clear maximum classes (in the aforementioned paragraph).

*2.3) The B&T subtree used for the observation has a very simple topology of only two branches. That might limit the generality of the conclusion.*

The B and T lineage tree is certainly simpler than some systems, but it contains a combination of lineage decisions as well as lineage progressions. While we were restricted by the relative paucity of well-verified lineage trees, our algorithm has been successfully tested on a) microarray data from early hematopoietic progenitors, b) single-cell gene expression data from intestinal cell differentiation (added in the revised manuscript, now in Figure 3—figure supplement 2), c) single-cell gene expression data from early germ layer differentiation (accompanying Jang et al. manuscript). The predictions in this paper regarding a forebrain/hindbrain split in neural development have been experimentally verified with viral barcoding in human brain development (Yao et al., 2017). Together, this body of evidence suggests to us that the motif we identified and the approach we developed may have more general applicability.

*Indeed, the reconstructed lineage tree of the progenitors include a loop, which is not supported by the literature. Authors should discuss this difference from known tree topology.*

The loop we predict in the progenitor tree is in fact supported by the Adolfsson model in the literature (Adolfsson et al., 2005). We have written a discussion of the differences between the reconstructed lineage tree and the traditional model (subsection “A lineage tree for early hematopoiesis”, second paragraph). To address the conflicting models of early hematopoietic progenitor lineages, we have compared the triplet topologies expected in the canonical model and in the Adolfsson model, and demonstrated that our predictions support the key features of the Adolfsson model (Figure 2—figure supplement 2).

*Also, why did they use their approach to reconstruct only hematopoietic progenitors and not all ~250 ImmGen populations?*

We did not test our model on the entirety of the ImmGen data set because not all of the lineage relationships were verified experimentally as the B- and T- cell branches and early progenitors, meaning the results would not necessarily have been informative of the performance of the algorithm.

*3) First algorithm.*

*3.1) For the formula in the third paragraph of the subsection “Using patterns to infer lineages”, but I wonder how to draw a cutoff for a tree to be 'correct'. If the method is applied to unrelated cell-types, what is distribution of the p-values?*

We thank the reviewers for this helpful suggestion. We compiled the inferred probability of triplet topologies for both related and unrelated triplets explored earlier from the B- and T- cell branches of the hematopoietic tree (see comment 2.2). We applied our Bayesian methodology to the set of 150 related and 100 unrelated triplets and compared the distributions of the probability of obtaining a non-null topology in a new supplemental subfigure (Figure 1—figure supplement 3). As mentioned in a previous comment and addressed in the new supplemental figure, the two distributions diverge considerably, with an AUC of 0.96 (Figure 1—figure supplement 3).

*3.2) How robust is the lineage prediction model to the initialization procedure and parameter choices? For example, how sensitive is the resulting tree on the number of initial clusters? how sensitive are the results for the algorithm's parameters (e.g., probability cutoffs) or for sub-sampling of samples or genes? A similar stability analysis should also be included for the second algorithm (extension to single cells), which can be even more sensitive due to its iterative nature.*

We have added a subsection “Stability analysis” to the Materials and methods section, covering both algorithms. We found that the resulting trees are robust to probability cutoff, the prior probability of having a null topology p(T), the relative probabilities of irrelevant and marker genes, and the number of initial clusters.

In a new subfigure (Figure 2—figure supplement 1), we have shown the distribution of maximal probabilities for non-null topologies in hematopoiesis. Here we show that triplets tend to have very high probabilities (close to 1), and varying the probability cutoff for triplet consideration does not affect the overall tree construction.

Additionally, a key input parameter into the algorithm is the expected prior distribution of means and standard deviations 𝑝(𝜇, 𝜎) used in the numeric integration (LXYZ). We have been using a uniform prior for both mean and standard deviation, but this requires specification of the ranges of the mean and standard deviation. We implemented an empirical prior 𝑝(𝜇, 𝜎) by estimating the empirical distribution over all the Immgen cell types, using kernel density estimation. The resulting hematopoietic lineage tree was identical. Using the kernel density estimated empirical prior may provide more stability in future analyses.

*4) Application to hematopoiesis:*

*4.1) It is not clear how well the algorithm can separate related from unrelated triplets? (i.e., representatives from three different branches). While this is mentioned anecdotally in Figure 2, it should be evaluated systematically, e.g., with AUROC.*

We have systematically annotated each of the 165 triplets, noting the topologies expected in the traditional model and in the Adolfsson model, and the expected triplet length in each model. This information has also been added to [Supplementary-material SD4-data]. and demonstrated that our predictions support the key features of the Adolfsson model (Figure 2—figure supplement 2).

Null topologies were identified by the algorithm for triplets with cells that are from 3 terminal nodes (e.g. the triplet MEP/GMP/FrBC) or from triplets that contain very distantly related triplets (e.g. LT/CLP/ETP).

For 7 triplets with terminal nodes in there is no expected topology (e.g. MEP/GMP/FrA), the method identified a non-null topology (e.g. GMP). However, these misidentified triplets did not affect the final tree, since they were eliminated using the pruning rule (subsection “Pruning rule”), which prunes triplets that contain links between cell types that are not nearest neighbors. In the case of MEP/GMP/FrA, MEP and GMP are separated by CMP, and GMP and FrA are separated by MLP.

By our count, the Adolfsson and traditional models differ in the topology of 9 triplets. The inferred expected topologies of 8 out of these 9 triplets support the Adolfsson model, which led to the identification of the final tree.

*4.2) Given the constraints of the algorithm, the aggregated lineage tree is undirected. While this is mentioned briefly, this key point should be better clarified in the main text.*

This is indeed a key point that is easily missed. We have amended the text to clarify this point, stating that “we next determined an undirected graph that recapitulates all of the individual triplet topologies (note that we are only inferring triplet topologies and are not inferring directionality).”

*5) Algorithm's extension to single cells.*

5.1) Comparison to other methods.

*Recently a number of papers have been published to infer cell lineages from single-cell gene expression data, such as Monocole (Trapnell et al. NBT, 2014), Treutlein et al. (Nature 2014), Wishbone (Setty, Nature Biotech, 2016), tSCAN (Ji, NAR, 2016), StemID (Grun, Cell Stem Cell 2016), and diffusion map (Haghverdi, Nature Methods, 2016). It will be useful to compare with these methods at least to get a sense to what degree this method provides new information (note: it looks like Figure 4 comes from Monocole, but we could not find a mention for it in the Results section).*

While our method is complementary to many of these alternative approaches, we have addressed some specific areas in which our method can add insight to analysis of single cell data. We applied Monocle 2, tSCAN, and StemID to the neuronal differentiation data set in this paper and were unable to extract meaningful results as shown in a new supplemental figure (Figure 4—figure supplement 2). In each of these cases, it was difficult to extract experimentally verified hypotheses from the analysis. In particular, the forebrain/hindbrain split in early development was not correctly identified (this split has been verified experimentally in Yao et al., 2017). We have also addressed this in the main text, adding that: “Analysis of this data with other recent methods such as Monocle and Monocle2 (Trapnell et al., 2014), TSCAN (Ji and Ji, 2016), and StemID (Grün et al., 2016) did not clearly reconstruct lineage or infer key genes regulating transitions (Figure 4 – Bottom, Figure 4—figure supplement 2). Monocle2 (Figure 4—figure supplement 2) produces a tree with complex branching, but *Sox2Sox2*^+^ progenitors and DCX+ differentiated neurons do no clearly separate”.

In addition, we performed a more in-depth comparison with StemID to explore the different conclusions one could gain from these orthogonal methods. We analyzed the intestinal cell differentiation system investigated by Grun et al., 2015 and compared our conclusions with those that they found. Using the Grun et al. cell clusters and a minimum cluster size of 10 cells, we accurately predicted the topology of each triplet in the lineage tree which they reconstruct, as shown in a new supplemental figure (Figure 3—figure supplement 3).

Our method is complementary to StemID in that for each of the triplets, we were able to identify a sparse set of transition genes which could be experimentally perturbed to explore the dynamics of the system ([Supplementary-material SD8-data]), many of which have known developmental relevance.

5.2) Differential expression in single cells.

*The t-test used to identify clear minimum and clear maximum patterns is appropriate for microarray analysis, but one limitation of t-test is it assumes the underlying distribution is normal, which is not true for RNAseq, and especially single-cell RNA-seq data. Specifically, single-cell RNA-seq can be severely under-powered because many genes (especially transcription factors) have low or even zero expression values due to dropout. Please discuss ways to incorporate more sophisticated tests that addresses this challenge (e.g., as in scDE), or demonstrate why this is not needed.*

We have now clarified that the single cell data is analyzed using the Bayesian formalism introduced earlier, and does not use a *t*-test to quantify separation between distributions. The inference methodology does, however, implicitly fit the expression data to a log-normal distribution. The choice of a log-normal distribution could potentially confound the results, particularly in cases with significant zero-inflation. This is a potential area for improvement in our algorithm, but the method can easily be adapted to model different distributions of RNA expression, such as γ distributions (Shahrezaei and Swain, 2008; Wills et al., 2013) and β-Poisson distributions (Vu et al., 2016; Delman and Hemberg, 2016). In either case, the probability of the data given different topologies would be computed by numeric integration over the parameters of the distribution, e.g. 𝛼 and 𝛽 for the Γ distribution, by replacing the log-normal distributions in [Disp-formula equ6 equ8 equ9] with ones from the appropriate model.

The choice of the right parametric form for single-cell RNA expression is still an area of active research. Our choice of log-normal distributions assumes that higher order moments of the distributions (beyond standard deviation) ought to have a minimal contribution to the predictions, but we have not tested this extensively.

*5.3) Related to that – the threshold used for definition of a pattern is p<0.005 in a two sample t-test. Though not specified by the authors, it seems that as there are 1500 transcription factors tested, and 3 tests performed per gene (to test the separation of each of the 3 triplet members from the other two), 4500 tests were performed per triplet. Thus, one would expect to find 22 transcription factors (4500*0.005) with a pattern that pass this threshold for each triplet in the absence of a signal. The authors have to address the multiple comparisons issue.*

We computed the FDR-adjusted p-values for the pattern recognition and found that the clear minimum pattern is still robust to this multiple hypothesis correction (Figure 1—figure supplement 2). In the Bayesian inference method, the prior odds of being a transition gene serves as a sparsity parameter which directly addresses the multiple comparisons issue.

*6) Application on Single cell data*

*6.1) While the single-cell data comes from another manuscript that is under review elsewhere; the information provided on this data is insufficient.*

*6.1.1) Please provide a compete "census" of the data – how many cells from each condition; how many reads per library, which read lengths were used?*

*6.1.2) How was the data processed? Should we worry about potential confounders of library quality (which have been presented and discussed time and again in the recent scRNA-seq literature)?*

We have added a new subsection to the Materials and methods to include this information (“In Vitro Neuronal Differentiation”). In particular, as the single cells were profiled independently using the Smart- Seq2 protocol (without pooling before amplification, as in methods such as Cel-Seq, STRT, or Drop-Seq), we did not observe the batch effects present in pooling-based methods. We would also like to note that the paper detailing the collection of this data set has now been published in Cell Stem Cell(Yao et al., 2017).

*6.2) The algorithm should have been demonstrated first on a single cell RNA-seq datasets where the true cell type transitions tree is known, such as one of the publicly available hematopoietic progenitors single cell RNA-seq datasets.*

We applied our method to the intestinal cell differentiation from Grun et al., (Figure 4—figure supplement 3) and found that our method correctly predicts the topology of each triplet based on the Grun et al. reconstructed lineage tree. Further, we identified key genes in many of the triplets with known developmental relevance. By reducing the system to a sparse measure of distances between cell types in the lineage tree, we can provide complimentary biological insight to existing methods which would allow for further experimental investigation. We have modified the main text in the last paragraph of the subsection “A lineage tree for early hematopoiesis”, to discuss this additional analysis.

*6.3) There is no estimate of how sparse is the resulting code – how many genes can be used to decipher the topology? Can they be identified ahead? Is it possible that any group of genes of similar size (~1500) can code for cell states and cell transitions with the same success as the transcription factors?*

To address this suggestion, we explored the sparsity of the code based on our reconstruction. We assembled 20 triplets from brain development that were originally inferred to be non-null and had a maximal leaf-to-leaf distance no greater than 5. For each triplet, we ranked genes based on their odds of being transition genes 𝒪*_i_*. We then attempted to infer the correct topology using only the N genes per triplet with the largest odds, and varied N between 1 and 50 (Figure 4—figure supplement 1). We find that with just 4 genes per triplet (80 total), we can correctly infer the topology of each of these triplets. Note that these genes are ranked by their odds of being transition genes, which is agnostic to the true topology of the triplet. Given assignments of cells to cell types (or clustering), these genes can be found without knowing the topology of the tree by computing the odds. We thank the reviewers for pointing out this relevant feature of the tree reconstruction. We have added discussion of the sparsity to the last paragraph of the subsection “Inferred lineage tree for human excitatory neuronal progenitors from in vitro single-cell data over 80 days of differentiation”.

*6.4) It is not clear how the tree was generated. Was it using the generic algorithm from the previous section, or by enforcing the temporal order of the samples? If it is the latter – please describe the procedure and explain why the generic algorithm does not work.*

We have now clarified in the main text (subsection “Inferred lineage tree for human excitatory neuronal progenitors from in vitro single-cell data over 80 days of differentiation”, sixth paragraph) that the tree was compiled using a) triplets with P>0.6, b) knowledge that *Sox2Sox2*^+^ cell types were progenitors, and DCX+ cell types were differentiated (so a DCX+ cell type could not transition to a *Sox2Sox2*^+^ cell type), c) using the pruning rule as described in the Materials and methods section. Using these rules, the tree was assembled by hand. A method for computationally generating a tree from a set of triplets would be a valuable extension of this work.